# Coverage and system efficiencies of insecticide-treated nets in Africa from 2000 to 2017

Samir Bhatt[1], Daniel J Weiss[1], Bonnie Mappin[1], Ursula Dalrymple[1], Ewan Cameron[1], Donal Bisanzio[1], David L Smith[1,2,3], Catherine L Moyes[1], Andrew J Tatem[3,4,5], Michael Lynch[6], Cristin A Fergus[6], Joshua Yukich[7], Adam Bennett[8], Thomas P Eisele[7], Jan Kolaczinski[9], Richard E Cibulskis[6], Simon I Hay[3,10,11], Peter W Gething[1]*

[1]Spatial Ecology and Epidemiology Group, Department of Zoology, University of Oxford, Oxford, United Kingdom; [2]Sanaria Institute of Global Health and Tropical Medicine, Rockville, United States; [3]Fogarty International Center, National Institutes of Health, Bethesda, United States; [4]Flowminder Foundation, Stockholm, Sweden; [5]Department of Geography and the Environment, University of Southampton, Southampton, United Kingdom; [6]Global Malaria Programme, World Health Organization, Geneva, Switzerland; [7]Center for Applied Malaria Research and Evaluation, Department of Global Health Systems and Development, Tulane University School of Public Health and Tropical Medicine, New Orleans, United States; [8]Malaria Elimination Initiative, Global Health Group, University of California, San Francisco, San Francisco, United States; [9]Strategy, Investment and Impact Division, The Global Fund to Fight AIDS, Tuberculosis and Malaria, Geneva, Switzerland; [10]Wellcome Trust Centre for Human Genetics, University of Oxford, Oxford, United Kingdom; [11]Institute for Health Metrics and Evaluation, University of Washington, Seattle, United States

*For correspondence: peter. gething@zoo.ox.ac.uk

**Abstract** Insecticide-treated nets (ITNs) for malaria control are widespread but coverage remains inadequate. We developed a Bayesian model using data from 102 national surveys, triangulated against delivery data and distribution reports, to generate year-by-year estimates of four ITN coverage indicators. We explored the impact of two potential 'inefficiencies': uneven net distribution among households and rapid rates of net loss from households. We estimated that, in 2013, 21% (17%–26%) of ITNs were over-allocated and this has worsened over time as overall net provision has increased. We estimated that rates of ITN loss from households are more rapid than previously thought, with 50% lost after 23 (20–28) months. We predict that the current estimate of 920 million additional ITNs required to achieve universal coverage would in reality yield a lower level of coverage (77% population access). By improving efficiency, however, the 920 million ITNs could yield population access as high as 95%.

## Introduction

Insecticide-treated nets (ITNs), which comprise conventional (cITNs) and long-lasting insecticidal nets (LLINs), are the single most widely used intervention for malaria control in Africa, proven to significantly reduce morbidity and mortality via direct protection and community-wide reductions in transmission (*Lim et al., 2011*; *Lengeler and Lengeler, 2004*; *Eisele et al., 2010*; *Killeen et al., 2007*).

**eLife digest** Malaria is a major cause of death in many parts of the world, especially in sub-Saharan Africa. Recently, there has been a renewed emphasis on using preventive measures to reduce the deaths and illnesses caused by malaria. Insecticide-treated nets are the most prominent preventive measure used in areas where malaria is particularly common. However, despite huge international efforts to send enough nets to the regions that need them, the processes of delivering and distributing the nets are inefficient. This problem is compounded by the fact that little information is available on how many nets people actually own and use within each country.'

Bhatt et al. have now created a mathematical model that describes the use and distribution of nets across Africa since 2000. This is based on data collected from national surveys and reports on the delivery and distribution of the nets. The model estimates that in 2013, only 43% of people at risk of malaria slept under a net. Furthermore, 21% of new nets were allocated to households that already had enough nets, an inefficiency that has worsened over the years. Nets are also lost from households much more rapidly than previously thought.

It's currently estimated that 920 million additional nets are required to ensure that everyone at risk from malaria in Africa is adequately protected. However, Bhatt et al.'s model suggests that given the current inefficiencies in net distribution, the extra nets would in reality protect a much smaller proportion of the population. Taking measures to more effectively target the nets to the households that need them could improve this coverage level to 95% of the population. The next challenge is to devise distribution strategies to send nets to where they are most needed.

The World Health Organization (WHO) promotes a target of universal coverage for all populations at risk with either ITNs or indoor residual spraying (IRS), with the former representing the primary vector control tool in nearly all endemic African countries (*WHO, 2013a*). The international community has invested billions of dollars in the provision of at least 700 million LLINs since 2004 (*WHO, 2013a*). While these investments have led to enormous scale up in population access to ITNs (*Noor et al., 2009*; *Monasch et al., 2004*), the target of universal coverage remains distant and millions of African households at risk remain unprotected (*WHO, 2013a*).

Bridging this gap is a key component of future strategies to reduce further the burden of malaria in Africa (*WHO, 2014*), and will require sustained commitment from donors, policy makers and national programmes. Central to these efforts is the capacity to monitor reliably current levels of ITN coverage in populations at risk and evaluate the systems that give rise to this coverage. This, in turn, enables progress towards international goals to be tracked and opportunities for efficiency gains to be identified. Such information is essential for evaluating the existing commodity and financing shortfalls and assessing future requirements if the target of universal coverage is to be achieved.

## Modelling coverage

To facilitate standardised and comparable monitoring of ITN coverage through time, WHO and the Roll Back Malaria Monitoring and Evaluation Reference Group (RBM-MERG) has over the past decade defined a series of indicators to capture two different aspects of ITN coverage: access and use. Gold standard measurements of these indicators are provided by nationally representative household surveys such as Demographic and Health Surveys (DHS) (*Measure, 2014*), Multiple Indicator Cluster Surveys (MICS) (*UNICEF, 2012*), and Malaria Indicator Surveys (MIS) (*RBM, 2014a*). These surveys are carried out relatively infrequently, however, meaning they cannot be used directly for evaluating year-on-year coverage trends or for generating timely estimates of continent-wide coverage levels. In contrast, programmatic data such as the number of ITNs delivered and distributed within countries, while not describing coverage directly, are available for most countries and years (*WHO, 2013a* ). In a 2009 study, Flaxman and colleagues (*Flaxman et al., 2010*) used a compartmental modelling approach to link these programmatic and survey data, generating annual estimates of the two ITN indicators recommended at that time on access (*% households with at least one ITN*) and use (*% children < 5 years old who slept under an ITN the previous night*).

Since that study, there has been increasing recognition that a richer set of indictors is required to identify the complex nature of ITN coverage (*Kilian et al., 2013*). An intra-household 'ownership gap' may exist whereby many households with *some* nets may not have *enough* for one net between two occupants (the recommended minimum level of protection (*WHO, 2013b*). Similarly, a 'usage gap' may exist whereby individuals with access to a net do not sleep under it. In response, the measurement of two additional indicators was recommended: *% households with at least one ITN for every two people* and *% population with access to an ITN within their household* (assuming each net was used by two people) (*RBM, UNICEF, WHO, 2013*; *RBM, 2011*). In addition, the indicator on usage was extended to include the entire population rather than only children under 5 years old. This updated set of four indicators, used individually and in combination, has the potential to provide a nuanced picture of ITN access and use patterns that can directly guide operational decision making (*Kilian et al., 2013*). To achieve this, there is a need to develop modelling frameworks to allow all four to be tracked through time.

## Evaluating efficiency

Countries have an ongoing struggle to maintain high LLIN coverage in the face of continuous loss of nets from households due to damage, repurposing, or movement away from target areas. In response, systems need to be responsive to emerging coverage gaps by ensuring nets are distributed to households that need them and avoiding over-allocation (i.e. distribution of nets to those that already have them). Together, the rate of net loss and the degree of over-allocation of new nets play a key role in determining how efficiently delivery to countries will translate into household coverage levels. These factors are not currently well understood but triangulation of survey and programmatic data allows new insights into both.

## Estimating future needs

The WHO define universal access to ITNs on the basis that two people can share one net. Using the working assumptions of a 3-year ITN lifespan and a 1.8 person-per-net ratio (one-between-two but allowing for odd-numbered households), a simple calculation yields an indicative estimate of 150 million new nets required each year to provide universal coverage to an African population at risk of around 810 million (*WHO, 2013a*). To support country planning and donor application processes (*RBM-HWG, 2014*), a more elaborate needs assessment approach has been developed by the RBM Harmonization Working Group (RBM-HWG) and implemented by 41 of the 47 endemic African countries (*RBM, 2014*; *Paintain et al., 2013*). The tool takes into account the size and structure of national target populations, a 1.8 person-per-net ratio for mass campaigns, additional routine distribution mechanisms employed by countries, and volumes of previously distributed nets and their likely rates of loss through time. Countries have used these inputs to calculate requirements for new nets to achieve national coverage targets, leading to an estimated continent-wide need for 920 million ITNs over the 2014–2017 period (approximately 230 million per year) (*RBM, 2014*). This tool provides a transparent, intuitive and standardised mechanism for comparing forecasted needs against current financing levels and identifying likely shortfalls. However, calculated needs are sensitive to assumptions about how a given volume of new nets will translate into population coverage, and inefficiencies in the system such as such as over-allocation and rate of net loss are not accounted for explicitly in the current needs assessment exercise.

The purpose of this study is to define a new dynamic modelling approach, triangulating all available data on ITN delivery, distribution and coverage in sub-Saharan Africa in order to (i) provide validated and data-driven time-series estimates for all four internationally recommended ITN indicators; (ii) explore and quantify different aspects of system efficiency and how these contribute to reduced coverage levels; and (iii) estimate future LLIN needs to achieve universal access by 2017 under different efficiency scenarios and how these compare to existing needs assessment estimates.

## Results

### Net stock estimates

*Figure 1A* summarises the main inputs to and outputs from the stock-and-flow model for LLINs when aggregated at the continental level. Some 718 million LLINs have been delivered across the 40

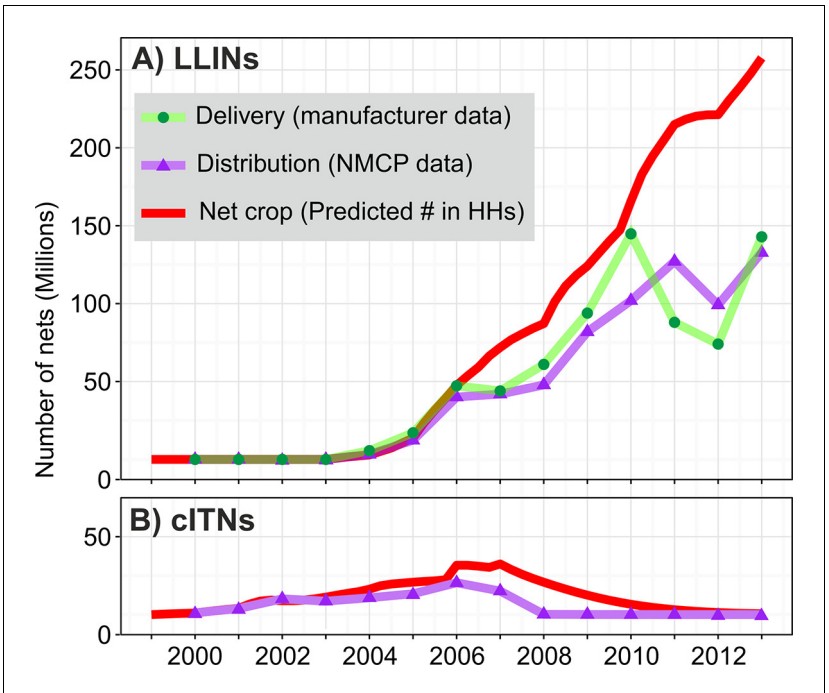

**Figure 1.** Time series of ITN delivery, distribution, and estimated net crop in sub-Saharan households 2000–2013 for (**A**) LLINs and (**B**) cITNs. Manufacturer data on deliveries were available for LLINs only. cITNs, conventional insecticide-treated nets; HHs, households; ITNs, insecticide-treated nets; LLINs, long-lasting insecticidal nets; NMCP, National Malaria Control Programme.

endemic countries since their introduction in 2004. As is well documented (*WHO, 2013a*), annual LLIN deliveries increased year-on-year from 2004 to 2010, reaching 145 million in that year, but then declined dramatically in 2011 and 2012 to less than half that amount before rising again to 143 million in 2013 (green line). Taking into account rates of loss in households, these LLIN deliveries led to a continental net crop shown by the red line. We estimate that there were 252 million LLINs in sub-Saharan households by the end of 2013, with that net crop growing approximately linearly from 2004, with the exception of a slow-down resulting from the reduced supply of nets in 2011–2012. *Figure 1B* shows equivalent distribution and resulting net crop estimates for cITNs, which constituted nearly all ITNs prior to 2005 but diminished rapidly in importance following the introduction of LLINs thereafter.

## Coverage estimates

*Figure 2* shows continent-level time-series estimates of the four internationally recommended ITN indicators, along with the 'access gap' indicator. All four indicators show a similar temporal trend: very low coverage levels and modest year-on-year increases for the first 5 years from 2000, with a marked inflexion point in 2005 and much more rapid gains thereafter. Importantly, however, all four indicators show that the pace of increase has, overall, slowed since 2005. By the end of 2013, we estimate that around two-thirds (66%, 95% CI 62%–71%) of households at risk owned at least one ITN. However, less than one-third (31%, 29%–34%) owned enough for one ITN between two people. This much lower level of adequate ownership is reflected in the levels of access and use, with 48% (45%–51%) of people at risk having access to an ITN within their household (on a one-between-two basis) and 43% (39%–46%) sleeping under an ITN the previous night. Comparison of *Figure 2A,B* demonstrates that many households that own *some* ITNs do not own *enough* for one-between-two, and this is captured in the time-series for the 'ownership gap' (*Figure 2E*). Encouragingly, this gap has been narrowed from 77% (76%–78%) of net-owning households having insufficient nets in 2000 to 56% (54%–57%) in 2013. Analysis of the 'use gap' suggested a large majority (89%, 84%–93%) of

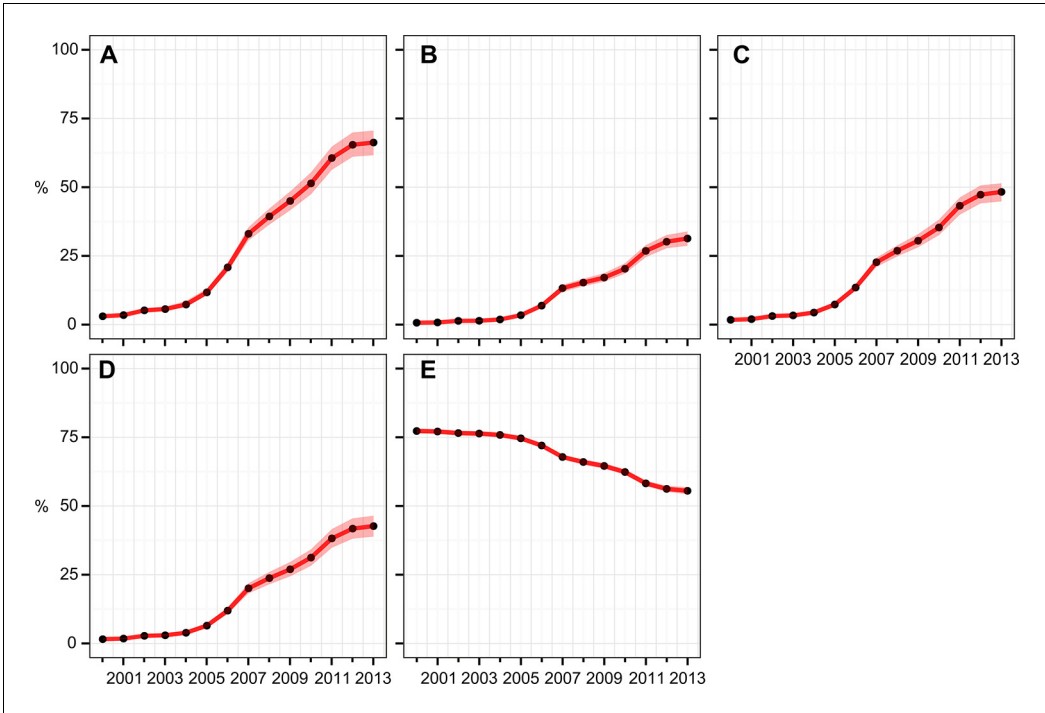

**Figure 2.** Continental-level time series of estimated ITN coverage indicators for the years 2000–2013. (**A**) % households with at least one ITN; (**B**) % households with at least one ITN for every two people; (**C**) % population with access to an ITN within their household; (**D**) % population who slept under an ITN the previous night; (**E**) 'ownership gap', the % of ITN-owning households with insufficient ITNs for one-between-two. Black circles are the annual estimates; pink envelopes denote the 95% posterior credible interval. ITNs, insecticide-treated nets.

those with access to an ITN in the household slept under it the previous night, and we found no evidence of significant change in this proportion through time.

The relatively smooth temporal trends seen at continental level obscure a great deal of complexity in the patterns of ITN scale-up occurring at national level (*Figure 3*). Nearly all countries began with very low coverage levels in 2000 and display a marked inflection point towards the middle of the decade, although there was considerable variation in the timing of onset of concerted scale-up activities. Importantly, the monotonic increases in coverage seen at the aggregated continental level are often replaced at national level with pronounced periods of rise and fall, and in many cases, 2013 does not represent the peak year. Variation in contemporary levels of coverage remains stark. The population with access to ITNs within the household, for example, was at or below 15% in seven countries in 2013, while above 70% for the top four.

## Over-allocation

Over the 14-year period since 2000, on average 15% (12%–18%) of all ITNs distributed to households were over-allocated (owned by households already owning sufficient nets for one-between-two). *Figure 4* illustrates how these over-allocation rates have changed through time. Around 7% (6%–9%) of ITNs were over-allocated in 2000, and this has risen steadily to 27% (22%–32%) in 2013. The year-on-year increase in over-allocation is to some extent an expected consequence of the overall growth in ITN provision: we found that over-allocation increased approximately 15 percentage points for each one-ITN-per-capita increase in net crop. Over-allocation also varied substantially between countries, for example ranging in 2013 from 50% (36%–65%) in the Republic of the Congo to 11% (9%–15%) in Côte D'Ivoire.

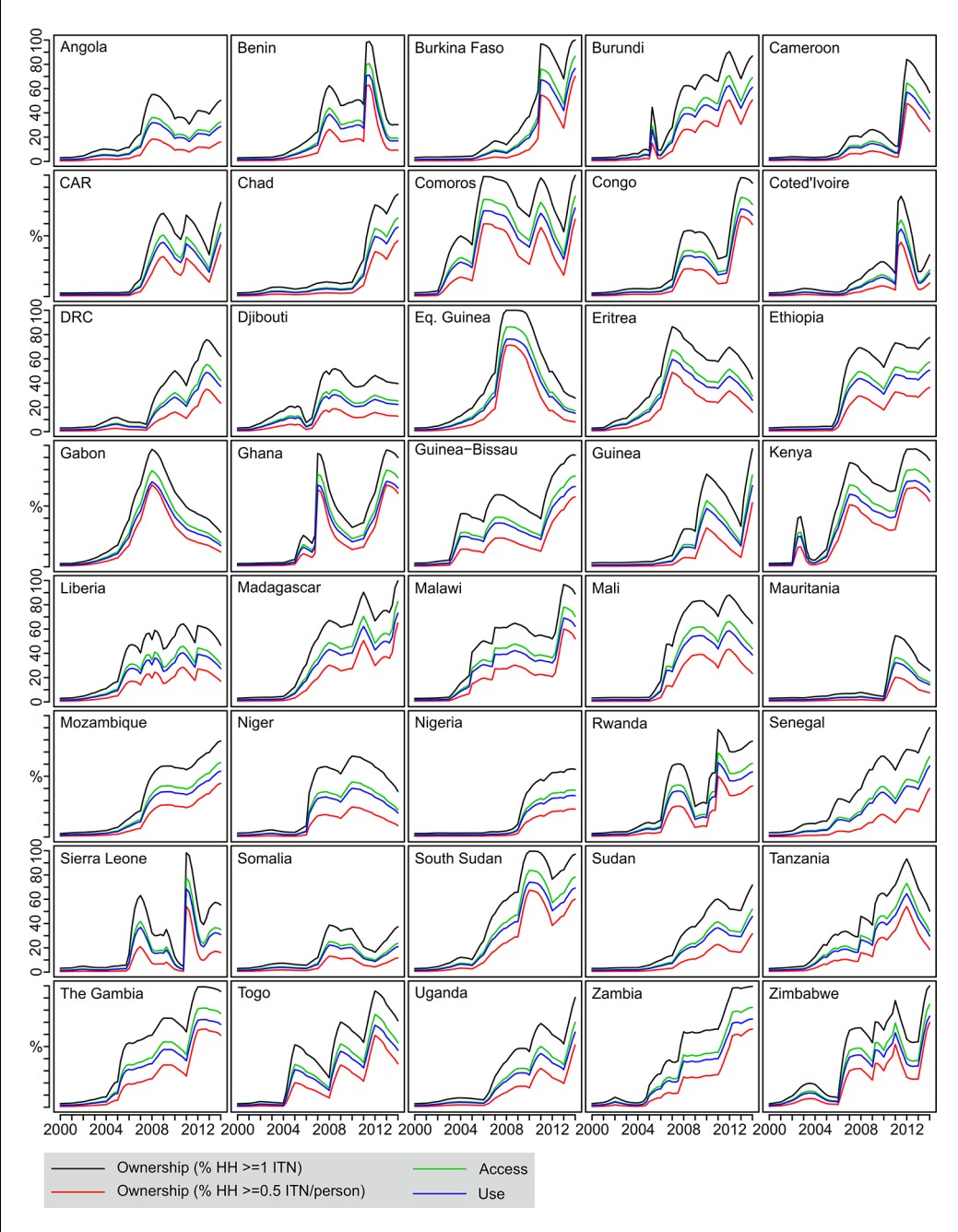

**Figure 3.** Country-level time series of estimated ITN coverage indicators 2000–2013. Each plot shows the four ITN coverage indicators: *% households with at least one ITN* (black); *% households with at least one ITN for every two people* (red); *% population with access to an ITN within their household* (green); *% population who slept under an ITN the previous night* (blue). CAR = Central African Republic; DRC = Democratic Republic of Congo; ITNs, insecticide-treated nets; HH = household.

## Net loss

Averaged over all years and all countries, we found the median retention time for LLINs in households was 23 (20–28) months. We found no statistically significant evidence of continent-wide temporal trends in retention times, but substantial between-country variation. *Figure 5* plots the LLIN loss function representing the most recent three years (2011–2013) for each country individually

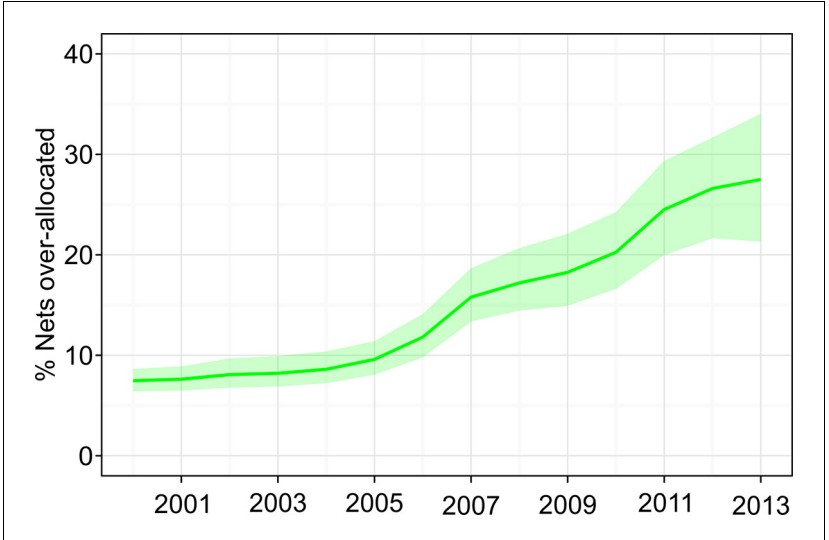

**Figure 4.** Time series of over-allocation for the combined set of 40 sub-Saharan endemic countries, 2000–2013. Over-allocation refers to insecticide-treated nets distributed to households already owning enough nets for one-between-two, measured as the percentage of over-allocated nets among all nets in households.

(blue lines), along with the aggregated continental-level curve (red line). For reference, we also over-lay on *Figure 5* some alternative loss functions that have been proposed. Flaxman *et al.* (orange line) fitted very small annual loss rates (5%) for years 1, 2 and 3 - with all LLINs then assumed lost after 3 years (*Flaxman et al., 2010*). The RBM-HWG proposed rate of loss (green line) is 8, 20 and 50% of LLINs to remain after 1, 2 and 3 years, respectively, with all nets being lost thereafter (*Net-works, 2014*). As can be seen, we found rates of loss for the first 3 years to be greater than both these alternatives for all countries. Both alternatives impose a three-year maximum retention time

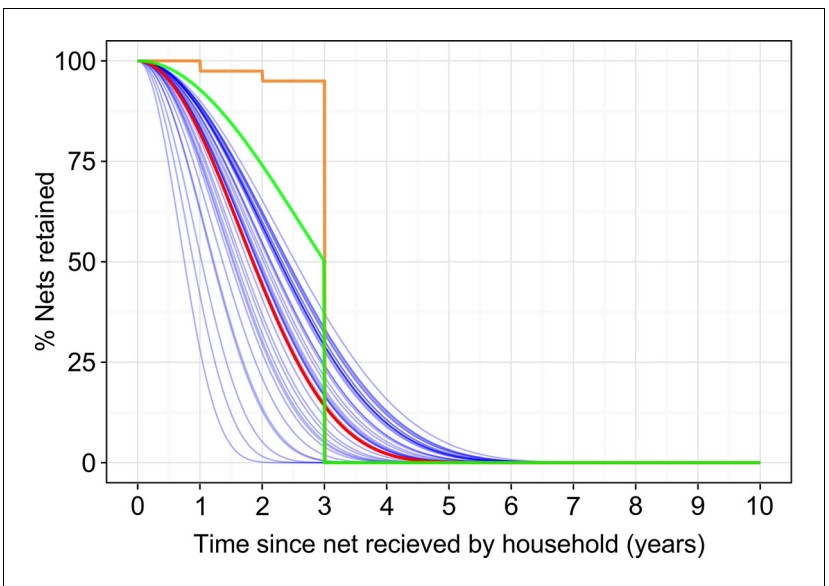

**Figure 5.** Insecticide treated net retention. Estimated long-lasting insecticidal net retention curves for each country individually (blue lines) and combined (red line), in both cases relating to the average of the most recent 3 years, 2011–2013. Also shown for reference are the rate of loss recommended in the Roll Back Malaria Harmonization Working Group needs assessment exercise (green line) and the loss rate fitted by Flaxman *et al.* (orange line).

and our decision not to do so meant that we modelled a small proportion of LLINs lasting some years beyond that point.

## ITN requirements to achieve universal coverage

*Figure 6* shows the projected levels of coverage that we estimate would be achieved by the end of 2017 with LLIN deliveries for the 2014–2017 period varying from zero to 2.5 billion and under a range of different efficiency scenarios. The most important characteristic of our results is the pronounced shallowing of the delivery-coverage curves: proportionately smaller gains are made in coverage as more LLINs are delivered in an archetypal 'law of diminishing marginal returns'. This means that under a business-as-usual scenario, where current levels of over-allocation and LLIN loss persist, very large increases in LLIN delivery would be required to achieve high coverage. Under this scenario, we estimate that 1 billion LLINs (i.e. an average of 250 million per year) would be required to achieve 80% of the population with access to an LLIN in the household by the end of 2017, although this would only translate into 70% population use.

The extent to which coverage gains diminish as deliveries increase is mitigated substantially when over-allocation and ITN loss rate are reduced. In a scenario with minimised over-allocation (where over allocation is set to zero), 80% population access in 2017 would be achievable with just 700

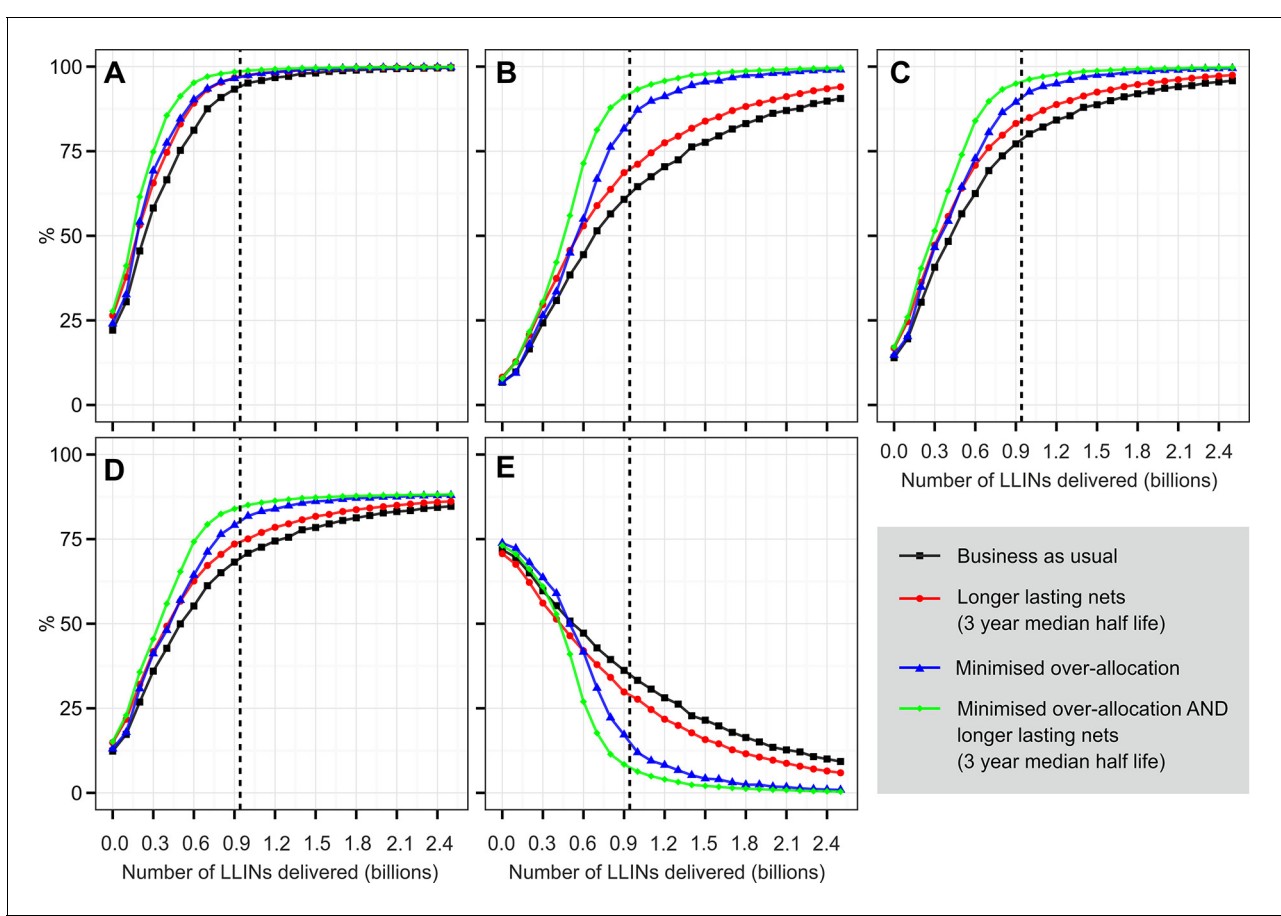

**Figure 6.** Projected 2017 coverage for sub-Saharan Africa in relation to number of LLINs delivered over 2014–2017 period. (**A**) *% households owning at least one ITN*; (**B**) *% households owning enough ITN for one between two*; (**C**) *% population with access to ITN within the household*; (**D**) *% population sleeping under an ITN the previous night*; (**E**) *'ownership gap', the % of ITN-owning households with insufficient ITNs for one-between-two.* For each indicator, we project likely coverage under four scenarios: current levels of over-allocation and net loss (i.e. 'business as usual'); with minimised over-allocation; with longer average net retention (3-yr median); and with both minimised over-allocation and longer net retention. The vertical dashed lines indicate the number of LLINs calculated as required over the period under the country programmatic needs assessment supported by Roll Back Malaria Harmonization Working Group. LLINs, long-lasting insecticidal nets; ITNs, insecticide-treated nets.

million nets (175 million per year). Reducing ITN loss rate to a 3-year median retention time would have a broadly similar impact, acting in isolation, to minimising over-allocation. If these two hypothetical efficiency gains were combined, however, 80% access could be reached in 2017 with around 560 million nets (140 million per year). We found that the relative importance of the over-allocation and LLIN loss rates changed as more LLINs were introduced. Increasing LLIN retention times was the most important factor at low levels of net delivery, but as more and more nets were provided, over-allocation became progressively more important. This is intuitive since it becomes increasingly difficult to avoid over-allocation as more households obtain adequate numbers of nets.

For reference, we also plot on *Figure 6* the 920 million additional LLINs calculated by countries as required for universal coverage of targeted populations by 2017 under the RBM-HWG needs assessment exercise. Under current levels of over-allocation and net loss, we estimate that by the end of 2017 this quantity of new LLINs would translate into 77% access (among those populations targeted by countries for ITN coverage) and, assuming current behaviour patterns continue, 68% sleeping under an ITN. Under the combined efficiency scenario with minimised over-allocation and 3-yr median ITN retention time, however, the 920 million nets would approach universal access (slightly over 95%).

## Discussion

By linking manufacturer, programme and national survey data using a conceptually simple model framework, the intention has been to provide a transparent and intuitive mechanism for tracking net crops and resulting household coverage that reflects the input data while simultaneously providing a range of insights about the system itself. In doing so we have been able to (i) provide a new approach for estimating past trends and contemporary levels of ITN coverage; (ii) explore the effects of uneven net distribution between households and the rates of net loss once in households; and (iii) use these insights to estimate how many LLINs are likely to be required to achieve different coverage targets in sub-Saharan Africa.

We have, for the first time, extended dynamic model-derived estimation of ITN coverage to all four internationally recognized indicators, along with the two 'gap' metrics. Our results reinforce a simple message: while gains in ITN coverage have been impressive, there remains an enormous challenge if the goal of universal access is to be achieved and sustained. The importance of the new expanded suite of indicators is also exemplified: while an encouraging two-thirds of households now own at least one ITN, less than half of these have enough to protect everyone who lives there. This ownership gap is narrowing but the disparity remains evident across nearly all countries. Conversely, there is little evidence that non-use of available nets contributes substantially to low coverage levels. We therefore reinforce earlier studies that suggest the overwhelming barrier to not sleeping under an ITN is lack of access rather than lack of use *WHO, 2013a*; *Eisele et al., 2009*; *2011*; *Koenker and Kilian, 2014*; *Koenker et al., 2014*). Of course, non-use may be important in certain local contexts, and finer-scale analysis can support identification of areas where behaviour change communication interventions may be appropriate to reduce it (*Kilian et al., 2013*).

We found substantial over-allocation of nets to households already owning a sufficient quantity, and that this became more pronounced as overall ownership levels increased through time. Mass distribution campaigns can, in principle, be designed to minimise over-allocation and maximise evenness of nets allocated to households strictly on the basis of households members and pre-existing nets. As other studies have highlighted, however, any possible commodity savings achieved by such strategies must be compared against the operational cost of these more complex distribution mechanisms (*Yukich et al., 2013*). What is certain is that over-allocation becomes a major barrier to achieving universal coverage when levels of ITN provision are high because most new incoming nets are simply leading to surpluses in many households, while elsewhere there remains a shortfall. This may have a disproportionately high public health impact if those surplus nets are concentrated in households at lowest risk. Wealthier, better educated and more urban households may be better placed to obtain available nets but are often located in regions of lower transmission (*Steketee and Eisele, 2009*; *Webster et al., 2005*). While beyond the scope of the present study, the approaches we have developed here could be extended to consider these issues of equity in coverage versus risk in more detail.

One of the most important observations in our study is that LLINs may be lost from households at a substantially faster rate than is currently assumed. Importantly, we assess loss by comparing total inputs to countries (from deliveries) to total numbers in households (net crop), and so we measure real losses rather than, for example, reallocation of nets between relatives (*Koenker et al., 2014*). Longer retention times of the sort observed in some local studies are not supported by the body of evidence we have provided by triangulating large-scale net distributions and household survey data. This more rapid loss rate has potentially important implications for existing guidelines. Current RBM guidance is for mass ITN campaigns to be conducted every 3 years, complemented by continuous distribution of nets via routine channels in order to maintain coverage levels between those campaigns. However, whatever levels of coverage are achieved by a given campaign, we estimate that one-half of the campaign nets distributed, on average, will not be present in households just 2 years later. Our coverage time-series for many countries suggest that routine distribution channels are not yet compensating fully for this rate of loss, often displaying pronounced dips in coverage levels between mass campaigns. Maintaining higher continuous coverage therefore clearly requires some combination of more frequent campaigns, greater ongoing distribution between campaigns, or more durable nets and improved care behaviour by users that lead to longer overall retention times.

We considered nets in households as simply present or absent, with no allowance for their condition. In reality, of course, nets may be retained by households (and thus 'present' in our calculations) even when they are badly torn, or have diminished insecticidal properties. As such, our estimates of 'coverage' would be revised downwards if additional measures of net efficacy were included. Our model is able to provide an estimate for every country and every year of the age-profile of ITNs in households. This raises the possibility of extending the predictions to incorporate modelled or observational data on average rates of net degradation in different contexts (*Briët et al., 2012*) to explore measures of entomologically effective coverage.

Tools developed to assist countries to calculate LLIN requirements, have tended to define need using a simple ratio to populations at risk (such as 1.8 people per net), and have made allowances for net loss from households using pre-defined rates of loss. We have been able to show that true LLIN requirements are likely to be considerably larger when the more rapid rates of loss are taken into account, along with the additional effect of likely over-allocation patterns. This more realistic framework not only provides the basis for more accurate needs assessments but identifies the relative importance of these different factors in determining the coverage that can be achieved for a given delivery level. Our analysis of future LLIN needs from the present time to 2017 demonstrates how these factors lead to a pronounced law of diminishing returns: as more nets are introduced to a population, proportional increases in coverage diminish, with over-allocation a particular problem at high net provision levels.

Under business-as-usual, the number of nets required to approach full coverage is prohibitively large. Clearly, however, reducing current system inefficiencies and increasing net retention are not straightforward and already the subject of much attention by countries and international partners. Over-allocation is the complex result of different distribution strategies and varying levels of population access to services, and any solution comes with its own cost. Net retention can doubtless be increased by improved LLIN technology coupled with behaviour-change communication efforts, although it is also feasible that retention times may reduce when overall net provision increases (with new nets displacing older ones). Additionally, we look only at the RBM definition of use and ignore the effectiveness of nets in repelling mosquitoes once they are being used. This is potentially an important confounder when considering retention times. While not aiming to provide solutions to these complex challenges, the results we present here provide an analytical framework in which the impact of theoretical efficiency gains can be assessed and this could be extended to include formal cost–effectiveness analysis.

In conclusion, our results provide evidence that LLIN requirements to achieve universal coverage have been underestimated. If obtaining higher coverage remains an accepted goal of the international community, then larger LLIN volumes must be considered and planned for at national and international levels. We emphasise, however, that this would be best achieved in parallel with a renewed focus on maximising the efficiency of coverage achieved for each new net financed. Given that the pattern of diminishing coverage returns for each dollar spent is likely to be unavoidable, the cost–effectiveness of pursuing universal coverage rather than a lower operational target must ultimately be weighed against alternative malaria control investments.

# Materials and methods

## Overview

Two important preceding studies have sought to model national-level ITN delivery, distribution, and coverage: the Flaxman et al. study (*Flaxman et al., 2010*) and the work of Albert Killian culminating in the NetCALC tool (*Networks, 2014*) and a series of related publications (*Paintain et al., 2013*; *Yukich et al., 2013*). Although very different in implementation, both approached the problem in a similar two-stage process. First, a mechanism was defined for estimating *net crop* — the total number of ITNs in households in a country at a given point in time—taking into account inputs to the system (e.g. deliveries of ITNs to a country) and outputs (e.g. the discard of worn ITNs from households). Second, empirical modelling was used to translate estimated net crops into resulting levels of coverage (e.g. access within households). We have adopted a similar analytical outline, but

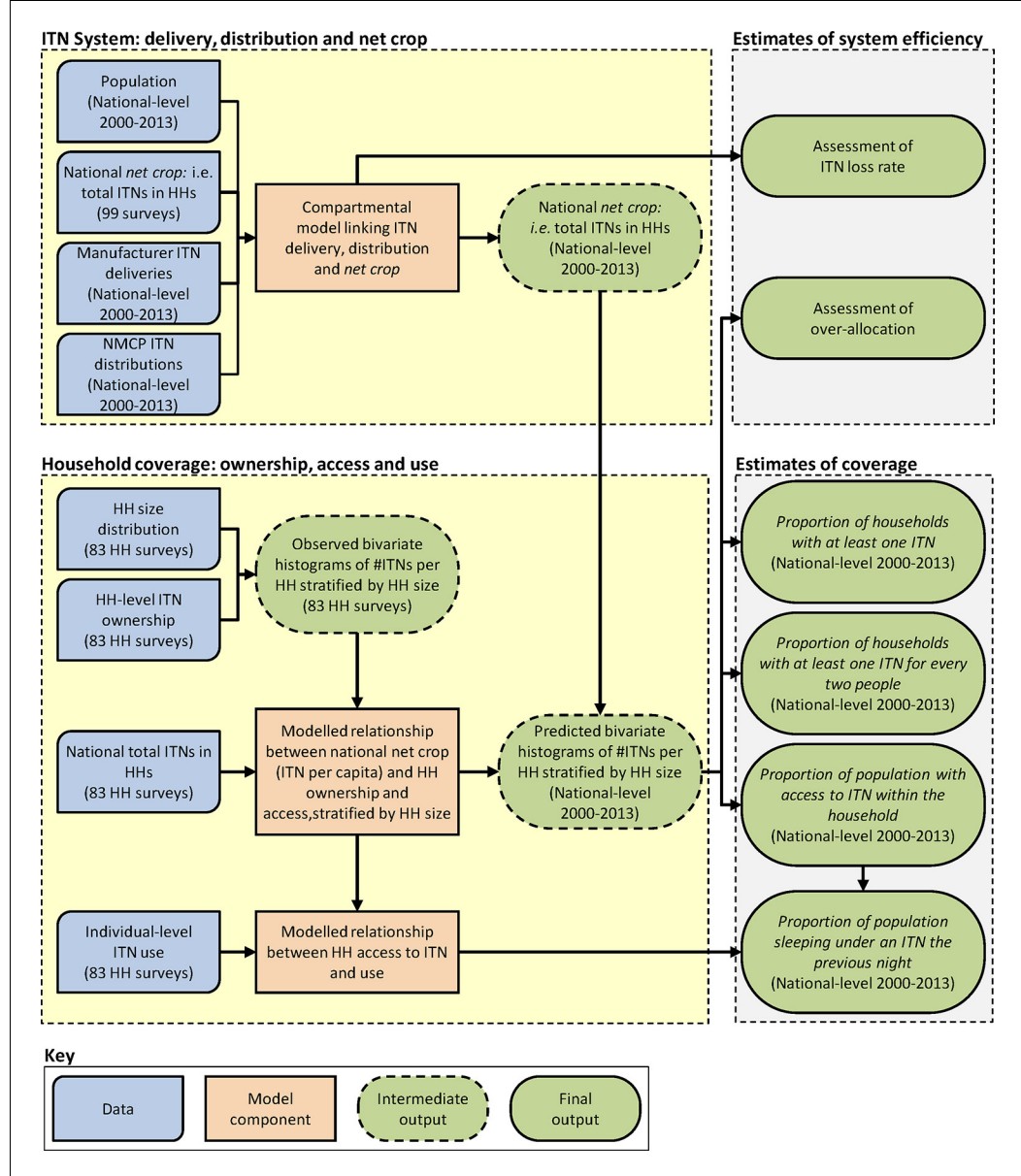

**Figure 7.** Schematic showing overall analytical framework linking data, model components, and outputs. HH = household; ITN, insecticide-treated net; NMCP = National Malaria Control Programme.

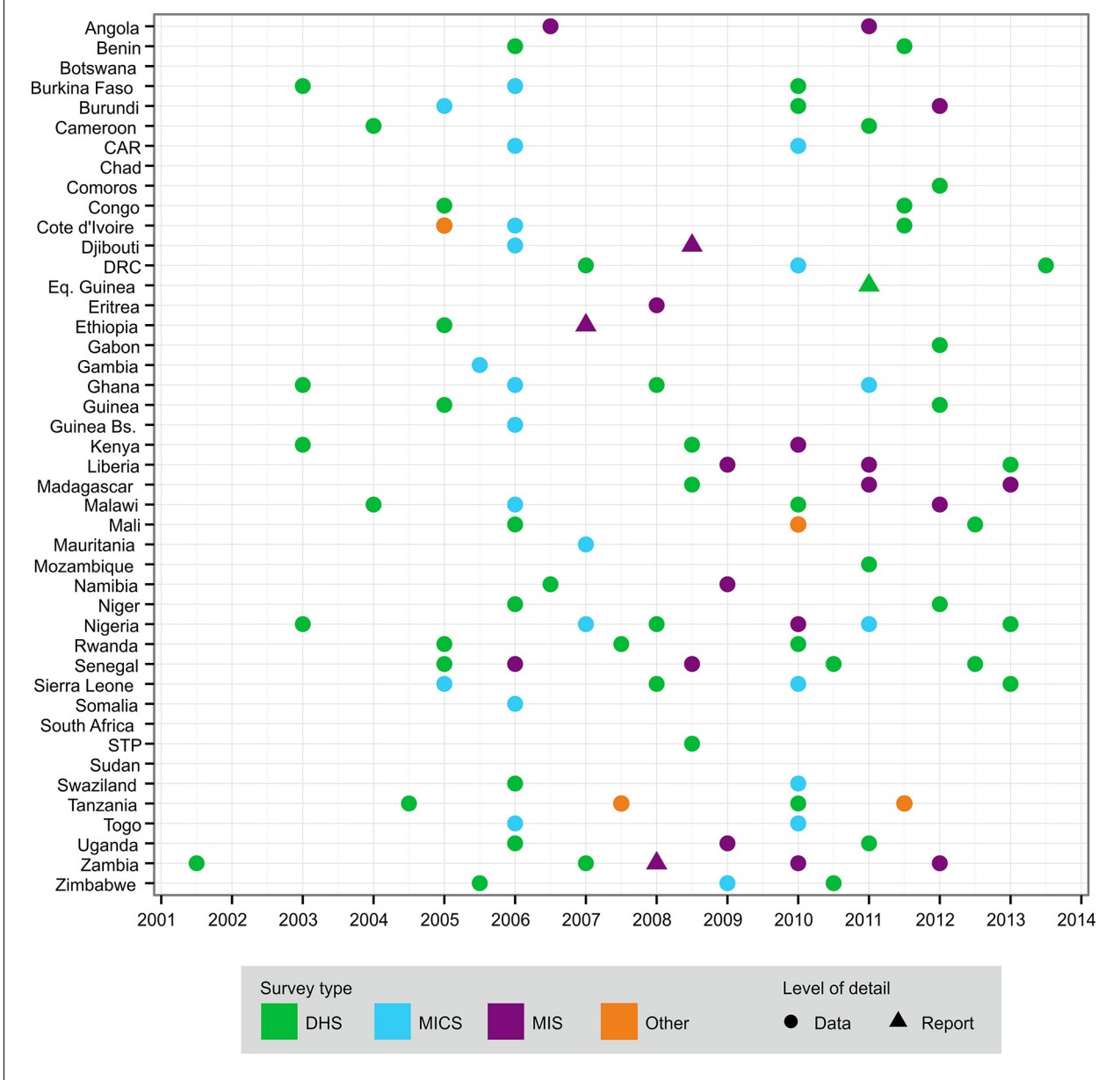

**Figure 8.** Distribution of national survey data on ITN net crop and household ownership, access and use used in this study, by country and year. The different types of survey are shown in the key: CAR = Central African Republic; DHS = Demographic and Health Survey; DRC = Democratic Republic of Congo; ITN, insecticide-treated net; MICS = Multiple Indicator Cluster Survey; MIS = Malaria Indicator Survey; STP = São Tomé and Príncipe.

the models we have developed for each stage differ structurally and conceptually from these earlier efforts. Our underlying principle has been to represent the ITN system in a simple and intuitive way and to parameterise that system using a data-driven approach that minimises the reliance on assumptions or small external datasets. In this Methods section, we describe: (i) the main data sources used; (ii) a new compartmental model for estimating net crop that also offers insights into rates of ITN loss from households; (iii) a new coverage model linking net crop to household net access and use that also assesses the efficiency of between-household distribution (i.e. the extent of over-allocation); and (iv) the use of our models to predict future ITN requirements to meet the goal of universal access. A schematic overview of our analytical framework is provided in *Figure 7*, and additional methodological detail is provided in the Supplementary Information.

## Data

We used three principal sources of data to fit our models. These are described briefly below and in more detail in Supplementary Information.

i. *LLINs delivered to countries:* data provided to WHO by Milliner Global Associates on the number of LLINs delivered by approved manufacturers to each country each year (*WHO, 2013a*; *AMP, 2014*). These were complete for each country from 2000 to 2013 inclusive.

ii. *ITNs distributed within countries*: data provided to WHO by National Malaria Control Programmes (NMCPs) on the number of cITN and LLINs distributed annually within each country (*WHO, 2013a*). Data were available for 365 of the 560 country-years addressed in the study. We treated these data as only partial records of distribution activities because the extent to which NMCP reporting captures distribution by non-government agencies is not known for all countries.

iii. *Nationally representative household surveys*. We assembled 99 national surveys from 39 sub-Saharan African countries from 2001–2013, covering 18% of all possible country-years since 2000 (*Figure 8*). More recent surveys provided household-level data on the number of cITNs, LLINs, people within each dwelling, and people sleeping under nets the previous night. RBM-MERG guidelines detail the conversion of these data into the standardised ITN indicators (*RBM, UNICEF, WHO, 2013*) and, in combination with national population data (*UNPD, 2012*), they can yield an estimate of national net crop (see Supplementary Information). Older surveys had less information: providing data on use but not ownership, for example, or for cITNs but not LLINs (see Supplementary Information). For most surveys (95/99), we were able to access the underlying data, while for the remaining four we used only the survey report.

## Countries and populations at risk

Our main analysis covered 40 of the 47 (*WHO, 2013a*) malaria endemic countries of sub-Saharan Africa. We excluded six endemic countries on the basis that ITNs do not form an important part of their vector control programme, as reported by the respective NMCPs to the African Leaders Malaria Alliance, ALMA (M. Renshaw, pers. comm. 3rd August 2014). These were Botswana, Cape Verde, Namibia, São Tomé and Príncipe, South Africa and Swaziland. We also excluded the small island nation of Mayotte, for which no ITN delivery or distribution data were available. We limited all analyses to those populations categorized as being at risk by NMCPs (*WHO, 2013a*). When interpreting NMCP distribution and household ownership data, we made the simplifying assumption that all reported ITNs were distributed among, and owned within, households situated in malaria endemic regions (*Burgert et al., 2012*). Additionally, we used data from African Leaders Malaria Alliance (ALMA) on the proportion of populations at risk targeted for ITNs versus IRS, and downscaled targeted populations at risk accordingly. It should be noted that restricting the distribution of ITNs to populations at risk makes the assumption that no ITNs are distributed to populations not at risk.

## Estimating national net crops through time

Like Flaxman et al.(*Flaxman et al., 2010*), we represented national ITN systems using a discrete time stock-and-flow model. In this structure, a series of compartments were defined that contained a given number of nets at each time-step, with possible movement of nets from one compartment to another between time-steps (see Supplementary Information). Nets delivered to a country by manufacturers were modelled as first entering a 'country stock' compartment (stored in-country but not yet distributed to households). Nets were then available from this stock for distribution to households by the NMCP or other distribution channels. Years where NMCP distributions were smaller than available country stock represented potential 'under-distribution', with nets left to stockpile rather than reaching households. However, because of the uncertainty associated with NMCP distribution data, these discrepancies could simply reflect under-reporting of distribution levels. To accommodate this uncertainty, we specified the number of nets distributed in a given year as a range, with all available country stock as one extreme (the maximum nets that could be delivered) and the NMCP-reported value (the assumed minimum distribution level) as the other.

New nets reaching households joined older nets remaining from earlier time-steps to constitute the total household net crop, with the duration of net retention by households described by a loss function. In this representation, the net crop simply reflected the differences over time between inputs to and outputs from households. This meant that distribution, net crop, and the loss function together formed a closed system: the three must triangulate exactly and knowledge of any two components allowed the third to be calculated directly. Flaxman et al. (*Flaxman et al., 2010*) assembled data from six studies on ITN durability and rates of loss. Using a loss function fitted to these data, however, they found that the three components tended not to triangulate: net crops observed in surveys were too small, given the data on nets distributed to households and their modelled rate of loss. Their interpretation was that the number of ITNs distributed each year may be systematically over-reported by NMCPs, and a 'bias parameter' was included in the model, adjusting downward the volume of nets entering households in each country compared with reported levels. As described above, we took a different approach: with no a priori expectation that NMCP distribution reports exaggerate distribution levels. Rather than fitting the loss function to a small external dataset, we fitted this function directly to the distribution and net crop data within the stock-and-flow model itself. Conceptually, this reflected the view that the 560 country-years of distribution data triangulated against the 102 survey-derived national net crop values represented a more impartial and data-driven way of inferring rates of loss than using limited data from local ITN retention studies. Loss functions were fitted on a country-by-country basis, allowed to vary through time, and defined separately for cITNs and LLINs. We compared these fitted loss functions to existing assumptions about rates of net loss from households. The stock-and-flow model was fitted using Bayesian inference and Markov chain Monte Carlo (MCMC), providing time-series estimates of national household net crop for cITNs and LLINs in each country along with evaluation of under-distribution, all with posterior credible intervals. A complete technical description is provided in the Supplementary Information.

## Estimating national ITN access and use indicators from net crop

Levels of ITN access within households depend not only on the total number of ITNs in a country (i.e. net crop), but on how those nets are distributed between households. In simple terms, a more even distribution yields a greater proportion of households owning nets than if those same nets are concentrated in fewer households. Many recent national surveys report the number of ITNs observed in each surveyed household. This allows, a histogram to be generated that summarises the net ownership pattern (i.e. the proportion of households with zero nets, one net, two nets and so on). By analysing such data from multiple surveys, previous studies have demonstrated that histograms for different countries vary in a broadly predictable way according to national net crop (*Flaxman et al., 2010*; *Yukich et al., 2013*). By representing these histograms using a formal statistical distribution (such as the negative binomial), and linking its parameters to net crop, predicted histograms can be generated for any country-year for which a net crop estimate is available (*Flaxman et al., 2010*; *Yukich et al., 2013*). These histograms, in turn, allow direct calculation of the first access coverage indicator (*% households owning one or more ITN*). We took the view that this approach—linking net crop to a statistical distribution, and using the distribution to calculate access indicators—is preferable to the alternative of regressing the access indicators against net crop directly. The latter approach, used in the NetCalc tool (*Networks, 2014*), is simpler but provides less direct insight into the patterns of between-household ITN distribution that ultimately link net crops to access levels.

One aspect that is known to strongly influence the relationship between net crop and household ownership distribution is the size of households found in different countries (*Networks, 2014*; *Yukich et al., 2013*), which varies greatly across sub-Saharan Africa (Swaziland, for example has an average household size of around three members, while in Senegal the average is nearly ten). Household size also, of course, determines whether a given number of owned nets will be sufficient to provide access to all residents. We extended earlier analyses (*Flaxman et al., 2010*; *Yukich et al., 2013*) to explicitly account for household size: using a bivariate (i.e. two- rather than one-dimensional) histogram model to link net crop to ownership distributions for each household size stratum (see Supplementary Information). We replaced the negative binomial distribution with a 2-d zero-truncated Poisson distribution and, for each household size stratum, fitted the distribution using two parameters: (i) the proportion of households with zero ITNs and (ii) the mean number of ITNs per ITN-owning household. Using the household-level data from 83 national surveys, we found that both

parameters were strongly related to national net crop, allowing bivariate histograms to be generated for every country-year that were closely representative of the true ITN ownership distribution.

Stratifying our analysis by household size had three important advantages over earlier approaches. First, the distribution of net ownership tended to vary substantially between households of different sizes within a given country and this variation would be missed if all households were considered together. Accounting for this enabled better fits to the data. This makes sense: all else being equal, larger households would be expected to own more nets than smaller ones and so distribution patterns would differ systematically. Second, the bivariate ownership histograms predicted for each country-year could be used to directly calculate all three indicators of household access. While a simple univariate histogram allows calculation of *% households with at least one ITN*, a bivariate histogram means the number of both ITNs and people in every household can be triangulated which, in turn, allows direct calculation of the two additional indicators: *% households with at least one ITN for every two people* and *% population with access to an ITN within their household*, along with the 'ownership gap' (see Supplementary Information). Linking these bivariate histograms to our annual net crop estimates for each country meant we could predict time-series of the access indicators at the national level from 2000–2013, with all parameters fitted in a Bayesian framework providing posterior credible intervals around each time-series. We also combined the country-level results to generate a set of continent-level indicator time-series, representing overall coverage levels among populations at risk in the 40 endemic countries. Third, the bivariate histograms allowed analysis of over-allocation: certain cells of the histogram represented households owning more ITNs than were required to achieve access on a one-between-two basis, and the proportion of the total net crop falling in this category was examined through time for every country.

We took a different approach for the final indicator, *% population who slept under an ITN the previous night*. ITN use is less directly linked to national net crop and is primarily determined by the availability of nets within households (*Eisele et al., 2009*). A total of 83 of the 102 national surveys contained data allowing the relationship to be explored between ITN use and each of the three access indicators with, perhaps unsurprisingly, *% population with access to an ITN within their household* displaying the largest correlation (adjusted $R^2$ = 0.96). We fitted this relationship across the 83 surveys using a simple Bayesian regression model (see Supplementary Information) and used it to predict time-series of the ITN use indicator for every country. The ratio of population use to access revealed the 'usage gap'—the fraction of the population with access to ITNs not using them—and between-country variation in this ratio was also explored.

## Estimating ITN requirements to achieve universal access

Our two-stage modelling framework represented the pathway from ITN delivery into countries through to resulting levels of net access and use in households. It also accounted for two potential factors that act to reduce access levels, and allowed these to be quantified through time for each country. Using this architecture, it was possible to simulate delivery of any hypothetical volume of ITNs to a given country over a given future time period, to predict the levels of access and use that would result, and to examine the impact of different amounts of over-allocation and net loss. The current needs assessment exercise that countries are undertaking (*RBM-HWG, 2014*; *RBM, 2014*) is designed to identify the number of LLINs required to achieve coverage targets by 2017. We used our model to estimate the levels of access likely to be achieved if these forecast LLIN commodity needs were met across the 2014–2017 period under a 'business as usual' scenario, that is, with current levels of over-allocation and net loss, and compared these predicted levels with the objective of universal access among target populations. We then generalized this experiment to predict the likely level of coverage (for all four indicators) achievable by 2017 under a broad spectrum of LLIN delivery levels, equivalent to a total for sub-Saharan Africa (two of the 40 endemic countries in our study did not participate in the RBM-HWG needs assessment exercise [Djibouti and Equatorial Guinea], and so our scenario analysis is based on the set of 38 remaining countries; to maintain comparability through time, we combined needs assessment data for mainland Tanzania and Zanzibar, and for Sudan and South Sudan) of between zero and 2.5 billion nets across the 4-year period. Further, we ran these simulations under four scenarios: (i) 'business-as-usual' (where current levels of over-allocation and net loss were maintained); (ii) with no over-allocation (new LLINs are distributed preferentially to those households with zero LLINs, then to those with less than one-between-two); (iii) with

reduced LLIN net loss by households (using a modelled 3-year median retention time); and (iv) with both no over-allocation and a 3-year median retention time.

## Acknowledgements

The authors acknowledge the valuable assistance from Melanie Renshaw in providing information from the Roll Back Malaria Harmonization Working Group Programmatic Gap Analysis and other guidance in the interpretation of our results. We thank members of the Roll Back Malaria Monitoring and Evaluation Reference Group (RBM-MERG) and the World Health Organization Surveillance Monitoring and Evaluation Technical expert Group (SME-TEG) for their feedback and suggestions. We thank Clara Burgert of the DHS Program for her assistance with DHS Survey access and interpretation.

## Additional information

### Competing interests

SIH: Reviewing editor, *eLife.* The other authors declare that no competing interests exist.

### Funding

| Funder | Grant reference number | Author |
|---|---|---|
| Medical Research Council | K00669X | Samir Bhatt<br>Peter W Gething |
| Bill and Melinda Gates Foundation | OPP1068048 | Daniel J Weiss<br>Bonnie Mappin<br>Ursula Dalrymple<br>Ewan Cameron<br>Donal Bisanzio<br>Peter W Gething |
| Foundation for the National Institutes of Health | U19AI089674 | David L Smith |
| Bill and Melinda Gates Foundation | OPP1110495 | David L Smith |
| Fogarty International Center | | Simon I Hay<br>David L Smith |
| Wellcome Trust | 091835 | Catherine L Moyes |
| World Health Organization | | Michael Lynch<br>Cristin A Fergus<br>Richard E Cibulskis |
| Global Fund to Fight AIDS, Tuberculosis and Malaria | | Jan Kolaczinski |
| Bill and Melinda Gates Foundation | OPP1106023 | Simon I Hay<br>Peter W Gething |
| Bill and Melinda Gates Foundation | OPP1119467 | Simon I Hay |
| Wellcome Trust | 095066 | Simon I Hay |
| Bill and Melinda Gates Foundation | OPP1093011 | Simon I Hay |
| Department for International Development | | Peter W Gething |

The funders had no role in study design, data collection and interpretation, or the decision to submit the work for publication.

### Author contributions

SB, PWG, Conception and design, Acquisition of data, Analysis and interpretation of data, Drafting or revising the article; DJW, EC, DB, DLS, CLM, TPE, REC, Analysis and interpretation of data,

Drafting or revising the article; BM, UD, AJT, JY, AB, SIH, Acquisition of data, Drafting or revising the article; ML, CAF, JK, Acquisition of data, Analysis and interpretation of data, Drafting or revising the article

## Author ORCIDs
Bonnie Mappin, http://orcid.org/0000-0002-1205-719X
David L Smith, http://orcid.org/0000-0003-4367-3849
Catherine L Moyes, http://orcid.org/0000-0002-8028-4079
Simon I Hay, http://orcid.org/0000-0002-0611-7272

## Additional files

### Supplementary files
• Supplementary file 1. Surveys used in model fitting, including type of data available. HH = households; CAR = Central African Republic; DRC = Democratic Republic of Congo; STP = São Tomé and Príncipe.

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

## Appendix 1. Supplementary information on data and methods

### 1.0 Outline of document

In this Supplementary Information document, we augment the main manuscript by providing additional details on the data and modelling architecture developed in this study to allow prediction of insecticide treated bednet coverage, system efficiency and future needs. In Section S2, we describe the acquisition and processing protocols for data from national household surveys, NMCPs and ITN manufacturers. In Section 3, we describe the Bayesian compartment model which estimates, at quarterly intervals, the total number of ITNs in households in each country (the *net crop*). In Section 4, we describe a second modelling stage that predicts national-level ITN coverage indicators as a function of predicted net crop. Finally, in Section 5, we describe the use of the modelling framework to predict future ITN coverage levels under a range of hypothetical ITN delivery volumes and efficiency scenarios.

### 2.0 Data collection

The various modelling stages were fitted to data from three principal sources.

#### 2.1 ITN manufacturer reports

Manufacturer reports provided information on the number of LLINs delivered to each country each year by international manufacturers. These data were provided to the WHO by Milliner Global Associates and were complete for each country from 2000 to 2013 inclusive.

#### 2.2 National malaria control program reports

NMCP reports provided information about the number of LLINs and cITNs distributed in a county within a given year. These data were provided to WHO by NMCPs and were available for 365 of the 560 country-years addressed in the study. We treated these data as only partial records of distribution activities because the extent to which NMCP reporting captures distribution by non-government agencies is not known for all countries.

#### 2.3 Household survey reports

We identified and obtained data for the ownership and use of ITNs from household surveys conducted in sub-Saharan countries since 2000, including DHS, MIS, MICS, AIDS indicator surveys (AIS) and a malaria and anaemia prevalence survey (EA & P) (*Supplementary file 1*). Data at the household level were acquired from 95 national surveys from 39 countries from 2001 to 2014. In addition, we acquired national level data from four household survey reports for which we were unable to obtain the household level dataset. The range and number of the household survey data collected is depicted in *Appendix Figure 7*. For those surveys where household-level data were available on the type of nets owned (see *Supplementary file 1*), the number of ITNs owned was determined by the sum of each ITN in the household. A net was considered an ITN if it was an LLIN, or a pre-treated net obtained within the past 12 months, or a net that has been soaked with insecticide within the past 12 months. ITNs were then subdivided into the two classes, LLINs and cITNs.

For the surveys where data on the type of net was only available for one net in each household (see *Supplementary file 1*), the overall survey-level proportion of total nets for each net classification (non-ITN, LLIN, cITN) was determined and multiplied by the number of nets in the surveyed households to estimate the number of LLINs and conventional ITNs owned by each surveyed households.

# 3.0 Compartment model

## 3.1 Introduction

This section describes our implementation of a Bayesian hierarchical model to impute, at quarter-annual intervals, the total number of insecticide-treated bed nets (which can be LLINs or cITNs) in a given country.

To achieve this goal, we built a model incorporating the three distinct sources of information on ITNs described in the preceding section: manufacturer reports on ITNs delivered to countries, NMCP reports on ITNs distributed to households within countries, and household survey data providing direct cross-sectional estimates of net crop in households at a given time point.

The key challenge in our model was linking the delivery and distribution data sources to the net crop measurements. *Figure 2* shows our schematic representation of the system and the evidence synthesis of these three data sources. In a given year, a given volume of nets are delivered to a country from manufacturers (Appendix figure 8, green arrows), giving rise to a (as yet undistributed) country stock (Appendix figure 8, orange arrow). Nets from this stock then may be distributed to households in that or subsequent years, as captured by NMCP distribution data (Appendix figure 8, blue arrows). To link the NMCP distributed nets to the direct estimates of the net crop existing within households, we needed to account for the rate of loss of nets from households. That is, once nets are distributed, they remain within households for a given length of time until they are discarded. By tracking the net loss from NMCP deliveries (Appendix figure 8, brown arrows), we were able to estimate the total number of nets in a country at a given time point (e.g. in 2008, when a household survey was conducted [ Appendix figure 8, red arrow] by summing the nets of all ages [Appendix figure 8, purple line]). The compartment model was therefore calibrated by parameterising the loss function in such a way that the net crop observed in each national survey was consistent with the known influx of nets to households (following manufacturer delivery and NMCP distribution) and an estimated rate of loss from households. Our loss function was modelled to be temporally varying for a given country and parameterises the proportion of nets discarded through time.

## 3.2 Model specification

### 3.2.1 *Observed data*

#### 3.2.1.1 Observations of net crop from household surveys

*Conceptual description*

Household survey reports provided a direct measure of how many ITNs, of any age, existed in households in a country over a sampling period. For use in fitting the compartment model, household-level survey data on ITN ownership needed to be translated into a representative estimate for the total number of nets in a given country. To accomplish this, we summarised each household survey into two variables: the average number of LLINs and cITNs per household and the average household size (i.e. number of residents). The total number of ITNs was then calculated as the product of the population of the country at the time of the survey and the estimated ITNs per-capita as observed in the survey data, taking into account the survey weighting to ensure the arithmetic mean was nationally representative. The calculation of the total number of nets was completed by specifying probability distributions that allowed propagation of uncertainty from the household surveys.

*Formal description*

To estimate national *net crop* (total ITNs in households) from each national survey, three summary statistics were required:

a) The average household size, $h$ $h_{\mu.}$, with an associated standard error, $hh_{\sigma}^{2}$.

b) The average number of LLINs per household, $AvgLLIN_{\mu'}$, with an associated standard error, $AvgLLIN_{\sigma}^2$.

c) The average number of cITNs per household, $AvgcITN_{\mu'}$, with an associated standard error, $AvgcITN_{\sigma}^2$.

It should be noted here that, throughout this Supplementary document, we use the term standard error when referring to the standard deviation of the sample mean or the variability of an estimator.

Survey data came from four different sources—DHS Program, MICS4, MICS3 and other surveys for which only the published report (and not underlying data) were available, mainly MIS undertaken unilaterally by NMCPs or supporting partners. Different protocols were required to obtain summary statistics (a), (b), and (c) above for each survey type.

*DHS and MICS4 data:* These household survey reports provided direct data about the number of LLINs and cITNs per household and the household size. Therefore, means across all households were calculated using a weighted average that incorporated the published survey sample weights and the standard errors obtained through Taylor linearization (**Kish and sampling, 1965**).

*MICS3 data:* These household surveys provided direct data about the household size and number of bed nets per household, but did not provide information on the type of each net observed (e.g. LLINs, cITNs or untreated bet nets). The surveys did, however, provide a full description of *a single* net within each household. Therefore, across all such single nets in each survey, we determined the mean proportion of nets that were LLINs, cITNs or untreated nets along with the standard errors. We then multiplied these proportions with the computed average number of nets (of any kind) to determine the average number of LLINs and cITNs per household. Propagation of uncertainty was achieved using MCMC sampling (**Plummer, 2003**; **Gelman et al., 2013**).

*Other reports:* These surveys contained no disaggregated household data and only reported the averages of household size and numbers of LLIN and cITN nets with no standard errors. We therefore assumed a small 1% error on these estimates, consistent with the magnitude of sampling errors seen in other surveys.

Using metrics (a), (b) and (c) obtained from the 95 processed surveys, the total number of LLINs and associated standard error in a country reported by a survey were defined probabilistically as:

$$\mu_{SURVEY_{LLIN_{\bar{t},c}}} = \mathrm{E}[Population_{\bar{t},c}{}^{Normal}(AvgLLIN_{\mu}, AvgLLIN_{\sigma}^2)\big/{}_{Normal}(hh_{\mu}, hh_{\sigma}^2)] \quad (1)$$

$$\sigma_{SURVEY_{LLIN_{\bar{t},c}}} = \sqrt{\mathrm{VAR}[Population_{\bar{t},c}{}^{Normal}(AvgLLIN_{\mu}, AvgLLIN_{\sigma}^2)\big/{}_{Normal}(hh_{\mu}, hh_{\sigma}^2)]} \quad (2)$$

And similarly the total number of cITNs:

$$\mu_{SURVEY_{cITN_{\bar{t},c}}} = E\left[Population_{\bar{t},c}{}^{Normal}(AvgcITN_{\mu}, AvgcITN_{\sigma}^2)\big/{}_{Normal}(hh_{\mu}, hh_{\sigma}^2)\right] \quad (3)$$

$$\sigma_{SURVEY_{cITN_{\bar{t},c}}} = \sqrt{\mathrm{VAR}\left[Population_{\bar{t},c}{}^{Normal}(AvgcITN_{\mu}, AvgcITN_{\sigma}^2)\big/{}_{Normal}(hh_{\mu}, hh_{\sigma}^2)\right]} \quad (4)$$

Where $\mathrm{E}[\bullet]$ is the expected value, $\sqrt{\mathrm{VAR}[\bullet]}$ is the standard error, *c* is a given country and $\bar{t}$ is the mean sampling time of the survey. *Equations 1–4* $\mu_{SURVEY_{LLIN_{\bar{t},c}}}, \sigma_{SURVEY_{LLIN_{\bar{t},c}}}, \mu_{SURVEY_{cITN_{\bar{t},c}}}$ and $\sigma_{SURVEY_{cITN_{\bar{t},c}}}$ were therefore observed data inputs into the compartment model.

### 3.2.1.2 Manufacturer reports of LLIN deliveries

As shown in Appendix figure 8, manufacturer reports provided data on the number of LLINs delivered to a country in a given year. There were no corresponding reports for cITNs.

*Conceptual description*

The main purpose of the manufacturer data was to determine stock levels and 'cap' estimated NMCP distributions in each year (i.e. more nets could not be distributed than were potentially available in country stock). The manufacturer reports were complete (no missing values) and assumed to be of high fidelity. We therefore modelled the likelihood of the manufacturer data in a given country-year as a normally distributed random variable. The manufacturer reports did not report standard errors in the delivery numbers and therefore we assumed a uniformly distributed prior probability on the error. Because the manufacturer data served only to inform stock and cap NMCP distributions, no considerations of sub-annual timing were required.

*Formal description*

We define $Manufacturer_{c,t}$ (*observed data*) as the number of manufacturer LLIN nets delivered to a country $c$ at year $t$. The number of manufacturer LLIN nets sent ($\mu_{c,t}$) was modelled as:

$$Manufacturer_{c,t} \sim \textbf{Normal}(\mu_{c,t}, Manufacturer_{c,t}\sigma_{m,t}^2) \qquad (5)$$

With error $\sigma_{m,t} = \textbf{Uniform}(0, 0.075)0.3$

### 3.2.1.3 NMCP reports of ITN distributions

As shown in Appendix figure 8, NMCP reports provided information on the number of both LLIN and cITNs nets distributed in a given country $c$ at year $t$.

*Conceptual description*

As with the manufacturer data, the NMCP reports were assumed to be of reasonably high fidelity. Unfortunately, NMCP reports were not complete and contained missing values where countries failed to report the number of nets distributed in a year. To impute these missing values, we defined an informative prior probability distribution on the NMCP distributions for both LLINs and cITNs. We tested multiple different parameterisations of NMCP prior distributions, and evaluated performance of parameterisations using out-of-sample cross validation to choose the best model.

Our final choice of NMCP prior distributions was data-driven, using combined reports across all African country-years that had NMCP data. We scaled the reports to per-capita NMCP distributions to remove country-specific differences. We observed that this combined per-capita distribution approximately followed an exponential distribution with a zero inflation hurdle (to account for no deliveries). Additionally, when looking at the combined reports across time, it was clear that combined distribution varied temporally. We therefore modelled NMCP LLINs and cITNs separately for each year by disaggregating the continent wide per-capita distribution temporally into separate exponentials with zero inflation. Finally, to account for variability due to sample size, we fitted splines through these time series.

*Formal description*

We defined $NMCP\_LLIN_{c,t}$ (*observed data*), $NMCP\_ITN_{c,t}$ (*observed data*) and $NMCP\_TOTAL_{c,t}$ (*observed data*) as the number of NMCP LLIN nets, NMCP cITN nets and the sum of LLIN and ITN NMCP nets reported distributed in a country $c$ at time $t$.

As described above, the NMCP reports did not have standard errors and contained missing values. We therefore defined informative prior distributions on $NMCP\_LLIN_{c,t}$ and $NMCP\_cITN_{c,t}$. First, we defined two sets $\mathscr{L}_t$ and $\mathscr{I}_t$ as the combined set of per-capita rates of NMCP LLIN and cITN across all African countries at times $t$ respectively. The sets

$\mathscr{L}_T$ and $\mathscr{I}_T$ with $T \in [2000, 2001, \ldots, 2012]$ therefore contained all NMCP LLIN and cITN distributions across all country-years.

To characterise the hurdle exponential distribution, we used two parameters: the zero inflated components ($p_{LLIN}0_t$, $p_{cITN}0_t$), which defined the probability that in a given country-year no nets were distributed, and the exponential component ($p_{LLIN}1_t$, $p_{cITN}1_t$) which, given some distribution of nets took place, gave the per-capita rate of distribution (i.e. how many nets per-capita were distributed).

We therefore define the proportion of zero NMCP deliveries for LLINs and cITNs as:

$$p_{LLIN}0_t = \sum_{i \in \mathscr{L}_t} 1_{\mathscr{L}_{i,t}=0} \Bigg/ \sum_{i \in \mathscr{L}_t} 1_{\mathscr{L}_{i,t}} \tag{6}$$

$$p_{cITN}0_t = \sum_{i \in \mathscr{I}_t} 1_{\mathscr{I}_{i,t}=0} \Bigg/ \sum_{i \in \mathscr{I}_t} 1_{\mathscr{L}_{i,t}} \tag{7}$$

And the mean delivery rate at time $t$ as:

$$p_{LLIN}1_t = \sum_{i \in \mathscr{L}_t} \mathscr{L}_{i,t} 1_{\mathscr{L}_{i,t} \neq 0} \Bigg/ \sum_{i \in \mathscr{L}_t} 1_{\mathscr{L}_{i,t} \neq 0} \tag{8}$$

$$p_{cITN}1_t = \sum_{i \in \mathscr{I}_t} \mathscr{I}_{i,t} 1_{\mathscr{I}_{i,t} \neq 0} \Bigg/ \sum_{i \in \mathscr{I}_t} 1_{\mathscr{I}_{i,t} \neq 0} \tag{9}$$

Using these two parameters we then defined the prior distributions on NMCP LLIN and NMCP cITN as:

$$rate_{LLIN,t} \sim \begin{cases} \boldsymbol{Exponential}\left(\dfrac{1}{p_{LLIN}1_t}\right), \Omega_{LLIN,t} = 1 \\ 0, \; otherwise \end{cases} \tag{10}$$

$$rate_{cITN,t} \sim \begin{cases} \boldsymbol{Exponential}\left(\dfrac{1}{p_{cITN}1_t}\right), \Omega_{cITN,t} = 1 \\ 0, \; otherwise \end{cases} \tag{11}$$

Where:

$$\Omega_{LLIN,t} \sim \boldsymbol{Bernouli}(p_{LLIN}0_t) \tag{12}$$

$$\Omega_{cITN,t} \sim \boldsymbol{Bernouli}(p_{cITN}0_t) \tag{13}$$

The terms $rate_{LLIN,t}$ and $rate_{cITN,t}$ therefore characterised the prior distribution on the per-capita rates of NMCP LLIN and cITN distributions, respectively. The values of terms $rate_{LLIN,t}$ and $rate_{cITN,t}$ also contained missing values for some years and showed variability due to different sample sizes. Therefore, we fitted penalised regression splines through the time series for each parameter to create smooth prior parameters. The spline fitting was done using restricted maximum likelihood with rigorous selection done to find the optimal number of basis functions for the spline (see *Appendix figure 1* for the spline fits).

To form the likelihoods, these prior rates needed to be scaled by the population in a given country-year. Therefore, we defined terms $\delta_{c,t} = Population_{c,t}rate_{LLIN,t}$ and $\pi_{c,t} = Population_{c,t}rate_{cITN,t}$ as the final prior distribution on the NMCP distributions (not per-capita).

Using these prior distributions and allowing for some uncertainty, we model the likelihoods for the observed $NMCP\_LLIN_{c,t}$, $NMCP\_ITNc_{c,t}$ and $NMCP\_TOTAL_{c,t}$ as random normal variables with added error:

$$NMCP\_LLIN_{c,t} \sim \textbf{Normal}(\delta_{c,t}, \sigma^2_{d1,t}) \tag{14}$$

$$NMCP\_cITN_{c,t} \sim \textbf{Normal}(\pi_{c,t}, \sigma^2_{d2,t}) \tag{15}$$

$$NMCP\_TOTAL_{c,t} \sim \textbf{Normal}(\delta_{c,t} + \pi_{c,t}, \sigma^2_{d3,t}) \tag{16}$$

Where $\sigma_{d1,t} = \sigma_{d2,t} = \sigma_{d3,t} = Uniform(0, 0.01)$. Additionally, all distributions (**equations 14** and **16**) were zero truncated to prevent negative numbers of nets.

It should be noted that NMCP distribution numbers are not always accurate and potentially under- or over-estimate the number of nets distributed. This may result, for example, from nets being distributed through other sources and therefore not contributing to the total NMCP distribution sum. To account for this uncertainty, we allow the number of nets distributed to take a uniformly distributed value with a lower bound of the NMCP distribution and an upper bound of the number of nets able to be distributed (found from the net stock— see **Equation 22**).

### 3.2.2 *Compartment model structure*

Following the methods outlined above, we arrived at a set of observed data, with standard errors, of the number of nets delivered for all country-years (manufacturer), the number of nets distributed within-country for all country-year (NMCP) and sparse estimates of the number of nets of any age in households (i.e. net crop) in 95 country-years with available surveys

Our compartment model linked all these processes together by modelling four different processes. (i) Using delivery and distribution information to allow for LLIN net stock to accumulate, thereby allowing for more LLIN nets to be distributed than were delivered in a country-year. (ii) Disaggregating the total nets delivered in a country-year into quarter-yearly intervals to allow for a more realistic modelling of the temporal dynamics. (iii) Linking year-by-year distributions through a loss function that accounted for the rate of nets being lost from households after distribution as a function of the time from distribution. (iv) Calibrating the compartment model quarter-yearly estimates on the observed survey reports. These four modelled processes are now described in turn.

### 3.2.2.1 **Accumulation of LLIN net stock**

*Conceptual description*

Following the methodology described in Flaxman et al (2010) (**Flaxman et al., 2010**), we define a net stock variable $Stock_{c,t}$ for a given country-year. $Stock_{c,t}$ links together LLIN manufacturer deliveries and NMCP LLIN distributions by allowing for a surplus of nets to be built up in a country. Essentially, $Stock_{c,t}$ added a 'cap' on the number of nets NMCPs report to be delivered, and therefore created an upper bound on erroneously reported deliveries.

We also allowed for NMCP data to represent an under-estimate of true distribution levels. This could occur, for example, if the NMCP reporting system did not capture those nets being distributed by non-governmental agencies. To accommodate this uncertainty, we specified the number of nets distributed in a given year as a range, with all available country stock as one extreme (the maximum nets that could be delivered) and the NMCP-reported value (the

assumed minimum distribution level) as the other. It should be noted that due to the lack of manufacturer data for cITNs, this uncertainty was only incorporated for LLINs.

*Formal description*

First we define $\hat{\delta}_{c,t}$, the adjusted $\delta_{c,t}$, as the modelled parameter for the number of NMCP LLIN nets delivered in a country-year. We define $\hat{\delta}_{c,t_0}$ and $Stock_{c,t_0}$ at the first time point $t_0 = 2000$ as:

$$\delta_{c,t_0} \sim \begin{cases} \mu_{c,t_0}, & \delta_{c,t_0} > \mu_{c,t_0} \\ \delta_{c,t_0}, & Otherwise \end{cases} \tag{17}$$

$$\delta_{c,t_0}^{adj} \sim Uniform(\delta_{c,t_0}, \mu_{c,t_0}) \tag{18}$$

$$Stock_{c,t_0} = \mu_{c,t_0} - \delta_{c,t_0}^{adj} \tag{19}$$

And then define $\hat{\delta}_{c,t}$ and $Stock_{c,t}$ at subsequent time points recursively as:

$$\delta_{c,t} = \begin{cases} \mu_{c,t} + Stock_{c,t-1}, & \delta_{c,t} > \mu_{c,t} + Stock_{c,t-1} \\ \delta_{c,t}, & Otherwise \end{cases} \tag{20}$$

$$\delta_{c,t}^{adj} \sim Uniform(\delta_{c,t}, \mu_{c,t} + Stock_{c,t-1}) \tag{21}$$

$$Stock_{c,t} = Stock_{c,t-1} + \mu_{c,t} - \delta_{c,t}^{adj} \tag{22}$$

As shown in the above equations, if a country did not distribute as many nets as were delivered, stock levels can increase, but with a limit that a country cannot deliver more nets than stock permits. $\hat{\delta}_{c,t}$ has a probabilistic interpretation reflecting our uncertainty about whether the NMCP values reported the total number of LLINs that were able to be distributed or if the calibration of the stock and flow model on the survey data required more nets to be delivered. For cITNs, it was not possible to include a stock component in the compartment model as there were no manufacturer reports for cITNs.

### 3.2.2.2 Temporal disaggregation of NMCP LLIN and cITN deliveries

*Conceptual description*

Modelled variables $\hat{\delta}_{c,t}$ and $\pi_{c,t}$ defined the number of LLINs and cITNs distributed within a country-year. However, these variables were modelled on data that did not provide any information about when in the given year nets were distributed. This led to the potential for temporal inconsistency when calibrating survey estimates at an average time in a country-year against NMCP distribution information with no-sub annual temporal resolution. Therefore, we needed to specify a prior on when NMCP distributions occurred within a given country-year. In striving to keep the model as parsimonious as possible, we first modelled a scenario where *all* NMCP nets were distributed at either the start or end of the year. This, however, did not represent reality adequately, and led to poor calibrations with survey estimates in some instances. We then relaxed this assumption to allow *all* nets to be delivered at a random point in the year, but again this led to poor calibrations. Finally, we opted for a more realistic distribution scenario where we disaggregated distributions to a quarter- yearly temporal resolution. We then assigned priors on NMCP quarterly net distributions to allow any proportion of nets to be delivered at the start, first quarter, second quarter, third quarter or end of the year. This scheme allowed for maximum flexibility in the model with minimal subjective prior assumptions, and yielded excellent calibrations with survey estimates.

*Formal description*

We disaggregated modelled variables $\hat{\delta}_{c,t}$ and $\pi_{c,t}$ (the number of LLINs and cITNs distributed within a country-year) into intervals $Q \in [0.25, 0.5, 0.75, 1]$, representing the number of nets distributed by the first, second, and third quarter or end of a given year. We defined the proportions of $\delta_{c,t}$ and $\pi_{c,t}$ in each interval as:

$$g_i \sim \boldsymbol{Uniform}(0, 1) \tag{23}$$

$$q_i = {g_i} \big/ \sum_{j \in Q} g_j \tag{24}$$

Where $i \in Q$ and $\sum_{i \in Q} q_i = 1$ and of course $\sum_{i \in Q} \delta_{c,t} q_i = \hat{\delta}_{c,t}$ and $\sum_{i \in Q} \pi_{c,t} q_i = \pi_{c,t}$ i.e. the sum across the year is preserved.

### 3.2.2.3 Rate of net loss

*Conceptual description*

Our compartment model estimates NMCP distributions at quarterly intervals through time from manufacturer delivery data and the estimated stock accumulation. The final link to calibrate these quarterly distributions with survey observations of net crop is the rate of net loss. We model the net loss function as a smooth compactly supported function defined previously as part of the NetCALC tool (*Koenker et al., 2013*). We also model the loss function as non stationary in time, and represent this change through time using a moving average. By using a moving average as opposed to individual loss functions for each year, or quarter, we were able to learn temporal changes given the sparse data and not over represent the prior.

*Formal description*

We tried several different functional forms for net loss (Weibull, exponential, hill) and decided on specifying the form using a smooth-compact function defined previously (*Koenker et al., 2013*):

$$Loss(t, k, L) = \begin{cases} e^{k - k/(1 - (t/l)^2)}, & t < L \\ 0, & Otherwise \end{cases} \tag{25}$$

Where *k* and *L* are loss function parameters with $k, L > 0$. The smooth-compact loss function produced models with a the lowest information criteria (DIC) from the other forms and has been validated in previous studies (*Koenker et al., 2013*; *Yukich et al., 2013*).

For both LLINs and ITNs we use the same functional form in *Equation 25* but restrict the bounds on parameter *k* as uniform priors on this parameter produced strongly non-uniform functions (*Bornkamp, 2012*). Therefore, to achieve a diffuse uniform prior on the loss functions, we allowed *L* to vary within large bounds, thereby producing priors that allowed candidate loss functions with half-lives from 0.7 to 5 years. These priors were necessarily vague to allow for adequate flexibility in fitting country-specific loss functions.

$$k \sim \boldsymbol{Uniform}(16, 18) \\ L \sim \boldsymbol{Uniform}(4, 20.7) \tag{26}$$

To model the loss function through time, we define a moving average on parameters *k* and *L*. Therefore the moving average on the loss function for both LLINs and cITNs is defined as:

$$k_{mv,t} = \frac{k_t + k_{t-1} + \ldots + k_{t-(n-1)}}{n} \tag{27}$$

$$L_{mv,t} = \frac{L_t + L_{t-1} + \ldots + L_{t-(n-1)}}{n} \tag{28}$$

Where $n$ is the moving average lag and $t \in [2000, 2001, \ldots, 2013]$, and any terms with $t < 2000$ are ignored. From out-of-sample cross validation we found the optimal lag to be $n = 5$, i.e. a balance between over- and under-smoothing. It should be noted that $t$ is restricted to the range $2000 \leq t \leq 2013$, which is the range for which we have real data on NMCP reports, manufacturer reports and household surveys. For the future scenario predictions (described later), we assume any future net loss behaviour is the same as that occurring in 2013, that is:

$$k_{mv,t>2013} = k_{mv,t=2013} \tag{29}$$

$$L_{mv,t>2013} = L_{mv,t=2013} \tag{30}$$

### 3.2.2.4 Calibrating the compartment model against survey information

*Conceptual description*

Given the net distributions defined at quarterly intervals and the temporally varying loss function, a continuous prediction of the number of nets of any age in a country could be found at quarterly intervals simply by summing across nets of all ages for a given quarter. These quarterly predictions of the number of nets of any age were then calibrated in a likelihood of the observed survey estimates. In the presence of observed survey estimates, this likelihood helped 'learn' values for all the prior probability parameters outlined in the compartment model. In the absence of survey information, the model defaulted to the prior probabilities for all parameters and relied on the NMCP and Manufacturer reports.

*Formal description*

Consider the stock and flow model evaluated over a period 2000:2017, which yielded 73 quarterly intervals. Now consider two 73× 73 matrices labelled $M_{LLIN}$ and $M_{cITN}$. The rows and columns of these matrices represent the entire time period in quarterly intervals.

Consider the stock and flow model progressing column wise through these matrices, at year $t$ and quarter $\hat{t} \in Q$, the column and row index $ind = 4t + \hat{t}$ (e.g 2004.5 or year 5 quarter 3 would be index 23) stores $\delta_{c,t}q_{\hat{t}}$ and $\pi_{c,t}q_{\hat{t}}$ (LLINs and cITNs in year $t$ and quarter $\hat{t} \in Q$).

Then for each quarter after the distributions $\delta_{c,t}^{adj}q_{\hat{t}}$ and $\pi_{c,t}q_{\hat{t}}$, the remaining nets in subsequent quarters were filled row wise according to the loss function defined in *Equation 25*.

By summarising the stock and flow process in this manner, the total number of LLIN and cITN nets of all ages in a given quarterly time period was simply the column sums $\sum_{j=1}^{73} M_{LLIN,ind}$, $\sum_{j=1}^{73} M_{cITN,ind}$.

Finally, we calibrated estimates of the total number of nets of all ages, against those reported from the household survey reports:

$$\mu_{SURVEY_{LLIN_{\bar{t},c}}} \sim Normal(P_{LLIN}, \sigma_{SURVEY_{LLIN_{\bar{t},c}}}) \tag{31}$$

$$\mu_{SURVEY_{cITN_{\bar{t},c}}} \sim Normal(P_{cITN}, \sigma_{SURVEY_{cITN_{\bar{t},c}}}) \tag{32}$$

Where the normal standard deviation is given by those found from the survey reports (*Equations 2 and 4*). Additionally, $P_{LLIN}$ and $P_{cITN}$ was defined as a linear interpolation between the two closest quarters and the average survey time $\bar{t}$.

It should be noted that, when calculating the nets per-capita, two scalings were used: (1) Countries with the proportion of population at risk being less than 1 were scaled according to

the WHO-defined populations at risk proportion. (2) Countries partially dependant on IRS as a means of vector control were scaled as the proportion of the population at risk targeted with ITNs.

# 4.0 Indicators model

## 4.1 Introduction

Section 2 provides details on how, using yearly data on manufacturer deliveries and NMCP distributions calibrated using household survey reports, we estimated the number of LLINs and cITNs in households in each country-year at quarterly intervals. From these net crop estimates, we used population information to derive the nets per-capita for LLINs and cITNs. Standardized ITN coverage indicators (*Kilian et al., 2013*) were then estimated from net crop by leveraging the household survey information with the estimates of nets per capita to derive a set of indicators on net ownership and usage. These were:

- Indicator 1: *% households with at least one ITN*

- Indicator 2: *% households with at least one ITN for every two people*

- Indicator 3: *% population with access to an ITN within their household*

- Indicator 4: *% population who slept under an ITN the previous night*

- Indicator 5: *The ratio of population use to access or the 'ownership gap'*

## 4.2 Indicator model structure

Previous models attempting to evaluate total nets and nets per-capita have utilised negative binomial models unstratified by household size to estimate Indicator 1 (*Flaxman et al., 2010*). However, using these previous approaches, it was impossible to estimate Indicators 2–5. Here we introduce a new zero-truncated Poisson model stratified by household size, which has the ability to estimate all Indicators 1–5 with excellent precision.

To begin the model derivation, consider a household survey $\mathcal{H}$. Contained within $\mathcal{H}$ we were able to calculate a density/histogram of the number of households with a given number of ITNs (both LLINs and cITNs). *Appendix figure 2* summarises this density plot, and it is clear that Indicator 1 is trivially calculated (sum of the red bars divided by the total) from this histogram, but Indicators 2–5 are not. Previous modelling approaches (*Flaxman et al., 2010*) used this unstratified density and assigned a probability distribution (e.g negative binomial or Poisson) parameterised such that the observed density could be recreated using a small number of model parameters (for the negative binomial 2 and Poisson 1).

There are two key problems with this approach, first as highlighted above Indicators 2–5, which provide additional richness of information for decision makers, cannot be directly calculated from this one-dimensional histogram. Second, after experimenting with a large suite of probability distributions, we found that fitting two-dimensional summary histograms to household survey data often provided very poor fits. The key to these poor fits is the lack of stratification of the number of ITNs by household size, the absence of which ignores an important determinant on the number of ITNs per household. A more useful summary of $\mathcal{H}$ is the inclusions of a second dimension for household size (*Appendix figure 3*). From this two-dimensional model, it becomes possible to estimate Indicators 1–3 (*Appendix figure 4*). However, the problem remains: how do we recreate this two-dimensional density when we have no household survey information?

To accomplish this, we developed a model which, given a household size distribution, translates an estimate of nets per-capita (derived from the compartment model) into an accurate realisation of the three-dimensional histogram.

### 4.2.1 *Zero truncated poisson model*

The most logical model to recreate the three-dimensional densities for a given household strata is the Poisson distribution (or a negative binomial distribution for added over dispersion). However, we found that these models did not recreate the observed pattern accurately. We tried more complicated zero-inflated versions but these did not improve fits.

After looking across all 83 surveys for which we had all the relevant information to recreate the three-dimensional histograms, we realised the process to create the histograms had to be separated into two processes: (a) a process which, for a given household strata, gave the density of households with no nets ($P_0$), and (b) a process which, for a given household strata owning nets, gave the density of a given number of nets (1,2,3...) ($P_1$). To think about this intuitively, consider a process that first fills the zero category of ITNs per household in **Appendix figure 3**, and then fills the categories 1,2,3 etc.

Consider a household strata $h$ (e.g. households of size three persons) from $\mathscr{H}$, it is easy to calculate the proportion of households with no nets. This is the $P_{0,h}$ parameter. Of the remaining households owning one or more nets, we calculated $P_{1,h}$ as the average number of nets in a household strata. The most logical probability distribution to fill the densities given $P_{1,h}$ is again the Poisson distribution as $P_{1,h}$ is simply the mean of the Poisson.

However, because we have already filled the household with no nets density, the correct distribution is a zero truncated Poisson distribution. Unfortunately, the mean of the zero truncated Poisson distribution is now no longer just $P_{1,h}$ but $\mu = \lambda/(1 - e^\lambda)$, which does not have the same useful interpretation. Therefore, we solved (using a simple root finding) for the value of $\lambda$ that gave a zero truncated Poisson with the same mean as a standard Poisson with mean $P_{1,h}$, but excluded a zero category.

Using this model, parameterised by just two parameters per household strata $P_{0,h}$ and $P_{1,h}$, we evaluated Indicators 1–3 across the 83 relevant surveys with correlation values of more than 0.98 showing that the model reproduced the complex density pattern in the two-dimensional histograms with excellent accuracy.

### 4.2.2 *Zero truncated poisson model in the absence of survey information*

The zero truncated Poisson is a probabilistic distribution that translates $\mathrm{Noentity}$ into parameters $P_{0,h}$ and $P_{1,h}$ that were able to re-create the two-dimensional densities in **Appendix figure 4** from which Indicators 1–3 could be calculated. However, we still needed to estimate $P_{0,h}$ and $P_{1,h}$ for country years without survey information.

Given the logical dependence of $P_{0,h}$ and $P_{1,h}$ on the underlying nets per-capita, we created two functions which, given a household strata, $h$, translated nets per-capita, *npc*, into $P_{0,h}$ and $P_{1,h}$. i.e.:

$$f_0(npc, h) = P_{0,h} \quad \text{and} \quad f_1(npc, h) = P_{1,h} \tag{33}$$

After experimenting with non parametric spline models, we found that simple polynomial surfaces worked remarkably well and had the added benefits of computational efficiency and compatibility with the compartment model.

We divided household sizes into 10 sizes $(1, 2, 3, 4, 5, 6, 7, 8, 9, \geq 10)$ and then modelled $f_0(npc, h)$ as:

$$f_0(npc, h) = \alpha_0 + \beta_1 h + \beta_2 h^2 + \beta_3 npc + \beta_4 npc^2 + \beta_5 npc^3 \tag{34}$$

and $f_1(npc, h)$ as:

$$f_1(npc, h) = \alpha_{1,h} + \beta_{6,h} npc \tag{35}$$

The model for $f_0(npc, h)$ is therefore a two-dimensional surface that varies according to household size but the model for $f_1(npc, h)$ is a separate linear straight line function for each household category.

*Equations 34* and *35* were fitted using Bayesian linear regression with the uncertainty in the coefficients being propagated through the compartment model.

When performing 10-fold out-of sample cross validation (leaving out entire surveys), we found that $f_0(npc, h)$ predicted $P_{0,h}$ with a correlation of 0.98 and $f_1(npc, h)$ predicted $P_{1,h}$ with a correlation of 0.97, indicating extremely good fits.

Given these two functions, which parameterise the zero truncated Poisson model, we can calculate Indicators 1–3 from an estimate of nets per-capita for a country-year (whether we have a survey or not) from the resulting two-dimensional density.

### 4.2.3 *Estimating % population who slept under an ITN the previous night and the 'ownership gap'*

The proportion of people who slept under an ITN (Indicator 4) was highly correlated with the proportion of people with access to an ITN (the 'use gap', Indicator 5). Therefore, to evaluate Indicator 4, we used a simple linear relationship between access and use evaluated across all 83 surveys with the relevant information (see *Appendix figure 5*). Therefore, all that was required to evaluate Indicator 4 was to take Indicator 3 (which contained all the rich information about household size strata) and translate it through a linear relationship with noise:

$$Indicator\ 4 \sim \textbf{Normal}(0.8838889 * Indicator\ 3, 0.06258131\,\hat{}\,2 \tag{36}$$

Finally, Indicator 5 (the ownership gap) was calculated as 1–(Indicator 4/Indicator3).

### 4.2.4 *Additional note on household sizes*

It should be noted that one missing piece in this analysis is the distribution of household sizes for every country-year. This information does not exist and is very difficult to model. Therefore, we make two assumptions. First, while the populations are known to change over time, we assume that in the 13-year window of our analysis the distribution of household sizes stays constant within each country. This assumption is, to some degree, warranted as countries with serially sampled surveys showed extremely similar household size distribution patterns, and the resulting indicators do not change significantly if a different time point household size distribution is used. Second, for countries with no household size information (due to no relevant surveys), we use an average across all surveys.

## 5.0 Future predictions

Using the methods described in sections 2 and 3, we were also able to simulate the delivery of any volume of ITNs to a given country over a given future time period to predict the nets per-capita and full suite of indicators. Additionally, we were able to change the dynamics of this simulated future period to allow nets to be retained for a longer period (by varying the net loss function prior) and account for over-allocation of nets (where there is a skewed distribution of net distributions in households with some households having too many nets and some too few).

When simulating forwards in time from 2013, we made several assumptions:

1. No cITNs were distributed or delivered. This is a justifiable as, with the exception of Gabon in 2013, none of the 40 countries in our analysis delivered or distributed any ITNs in 2012 or 2013. Therefore, it is reasonable to assume that for future years, these countries continued using LLINs exclusively. This assumption also follows country recommendation by the WHO to distribute LLINs and not cITNs (*Measure, 2014*).

2. Because only projected information on manufacturer data is known, we assume that all nets delivered to a county are distributed. It should be noted that the long-term consequences of this assumption are insignificant as data from 2000–2013 show that nets that are delivered are ultimately distributed, however, short-term particulars of how many nets are distributed and how many are retained as stock are not captured.

3. Given a number of nets delivered to all of Africa during the period 2014–2017 (e.g. 500 million LLINs), we use the yearly proportions defined by the harmonisation working group (RBM-HWG) to determine how many nets a country gets in a given year. The working group estimates represent the most reliable projection estimates available.

4. If the loss function was chosen to be that fitted within the model, we assume the loss function for future years was fixed to that learnt for 2013. We justify this assumption by looking at the changes in loss function on fitted data from 2000–2013, which suggests that the loss function does not vary dramatically and is stable temporally.

## 5.1 Future scenarios

For the future scenarios the number of nets delivered to all of Africa during the period 2014–2017 was varied from 0 nets to 2.5 billion nets with increments of 100 million nets.

*Business as usual*

In the business-as-usual scenario, the loss function was fixed to that learned for 2013 and no accounting for over-allocation was performed.

*Minimised over-allocation*

In the minimised over-allocation scenario (where over allocation is set to zero), the loss function was fixed to that learned for 2013. To account for over-allocation, we ran the compartmental model to estimate net crop, as we would have under the business as usual scenario. However, when evaluating the coverage indicators, we implemented a 'redistribution' algorithm that operated on the modelled two-dimensional histogram, summarising the proportion of households of a given size having a given number of nets for each country-year. From this histogram, we removed those 'surplus' nets from households that that would normally occur due to over-distribution under business-as-usual. These were then redistributed to households with two few nets. This reallocation started with the biggest households with the largest shortfalls and, while surplus nets remained, continued into progressively smaller households until, potentially, every single household had sufficient nets for one-between-two.

*Longer net retention*

In the longer net retention scenario, we did not use the loss function fitted in the model, but rather defined a fixed loss function with a mean half-life of 3 years. The prior uncertainty was taken as the standard deviation across all actual fitted loss functions calculated retrospectively.

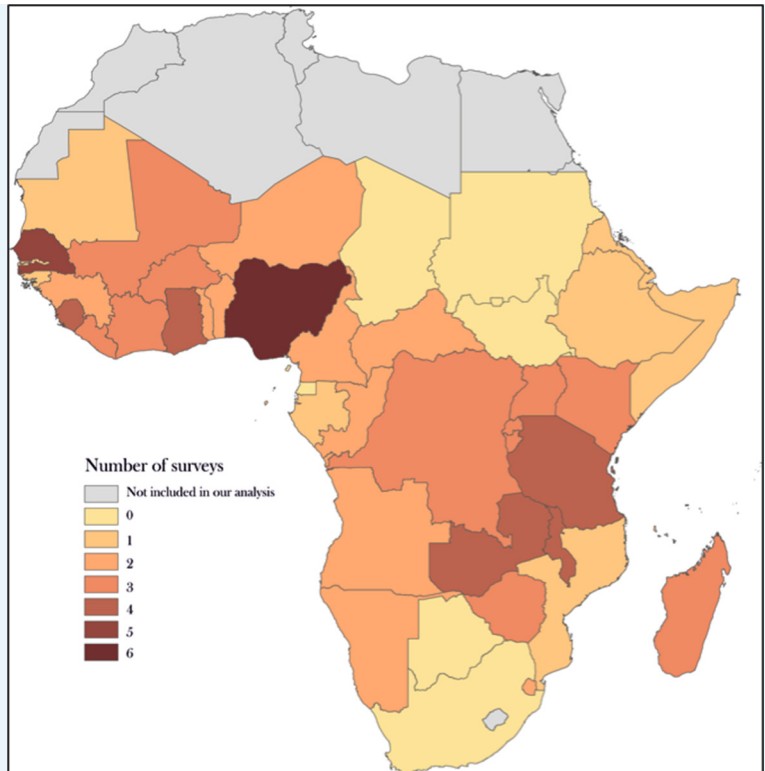

**Appendix figure 1.** Number of household surveys by country: Each country is shaded to repre-sent the number of household surveys datasets and reports acquired with data on insecticide-treated net ownership and use.

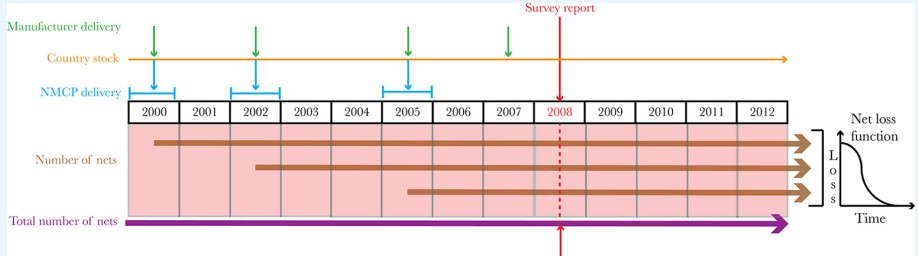

**Appendix figure 2**. Compartment model dynamics for LLINs. The compartment model predicts the total number of nets in a country (thick purple line). For a given country at given years there will be LLINs delivered to the country from manufacturers (green arrows), and LLINs distributed at some point in that or subsequent years within the country (blue arrows). Country-wide survey reports give a cross-sectional measure of the total number of nets in households (red line). Evidence synthesis of three data sources can then be inferred by estimating a rate of loss (via a loss function). LLINs, long-lasting insecticidal nets.

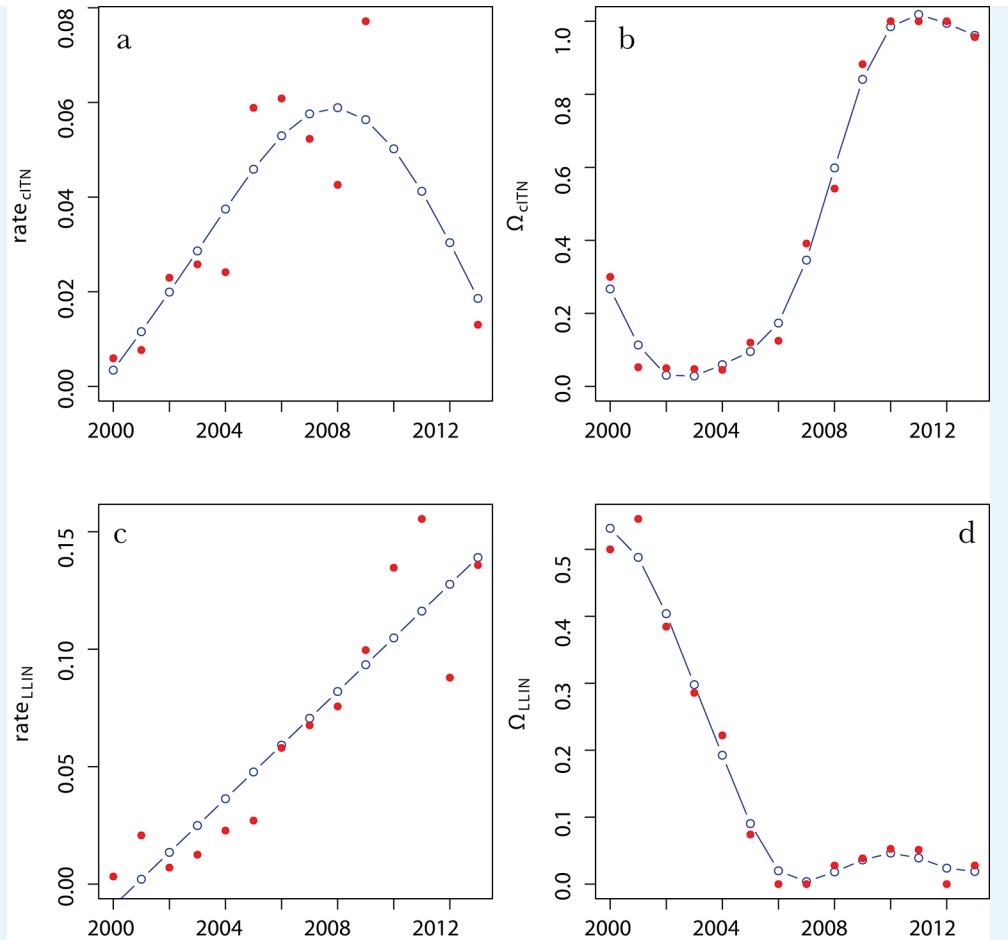

**Appendix figure 3**. Parameter fits for the hurdle exponential NMCP prior. (**A,C**) are the exponential rate priors on the per capita rates of cITN (**A**) and LLIN (**C**) distributions in a given year (see *Equations 10* and *11*). (**B,D**) are the Bernoulli hurdle probabilities that zero cITNs (**B**) or LLINs (**D**) are distributed in a given year (see *Equations 12* and *13*). Red points are the data, and blue line are optimised spline fits. cITNs, conventional insecticide-treated nets; LLINs, long-lasting insecticidal nets; NMCP, National Malaria Control Programme.

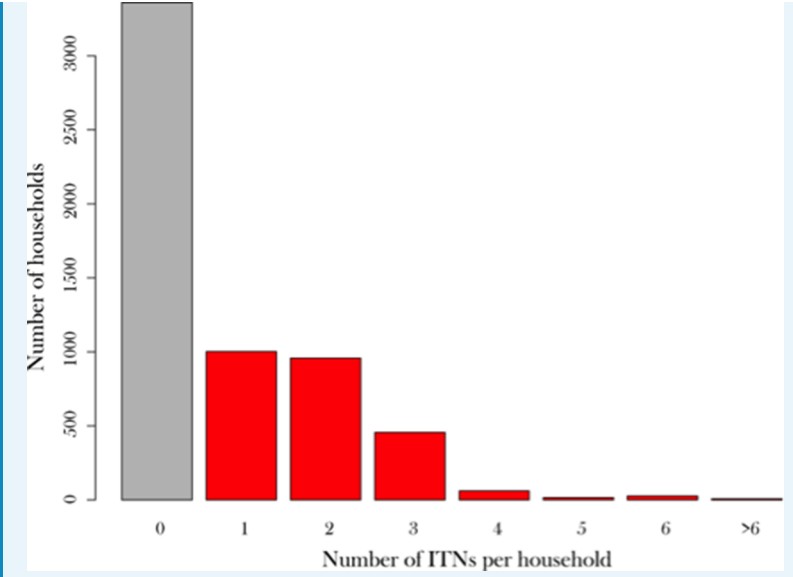

**Appendix figure 4**. Histogram of #ITNs per household as observed in the Nigeria 2010 Malaria Indicator Survey. Red bars represent the number of households with one or more nets, and the grey bar represents households with no nets. ITNs, insecticide-treated nets.

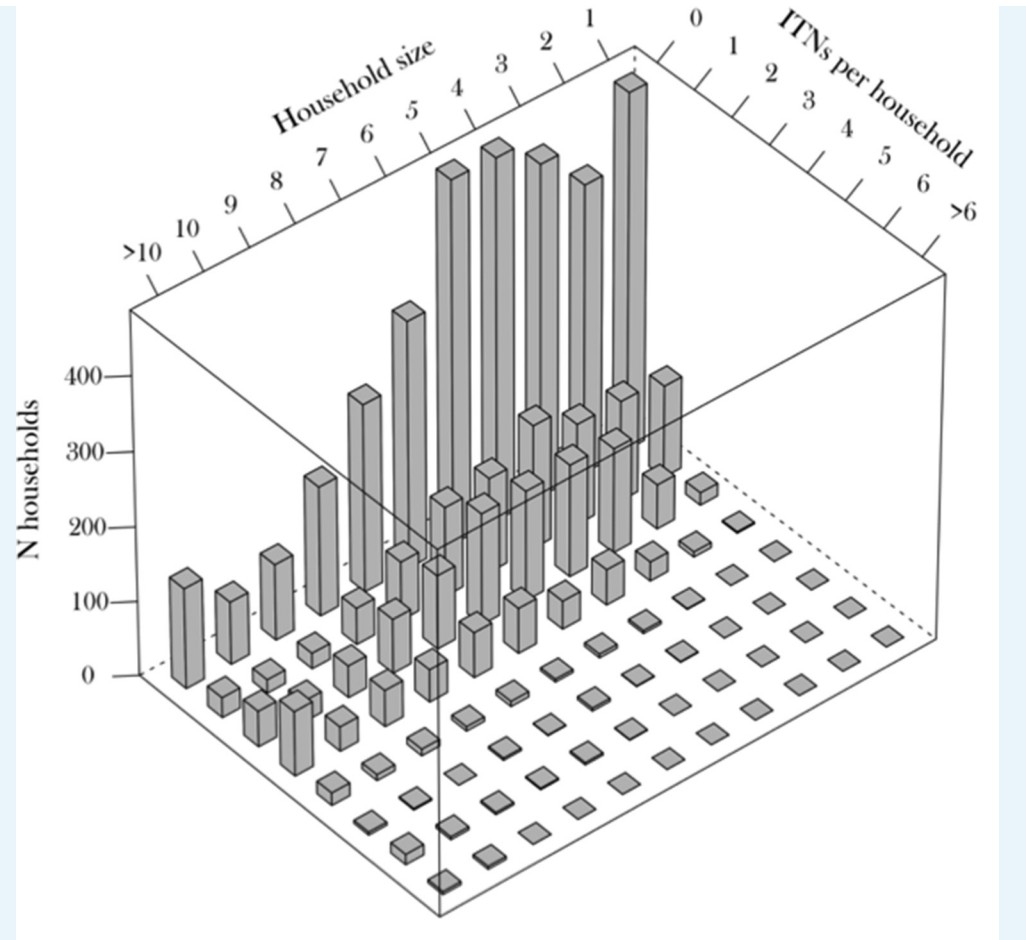

**Appendix figure 5**. Nigeria 2010 Malaria Indicator Survey household survey summary stratified by household size. ITNs, insecticide-treated nets.

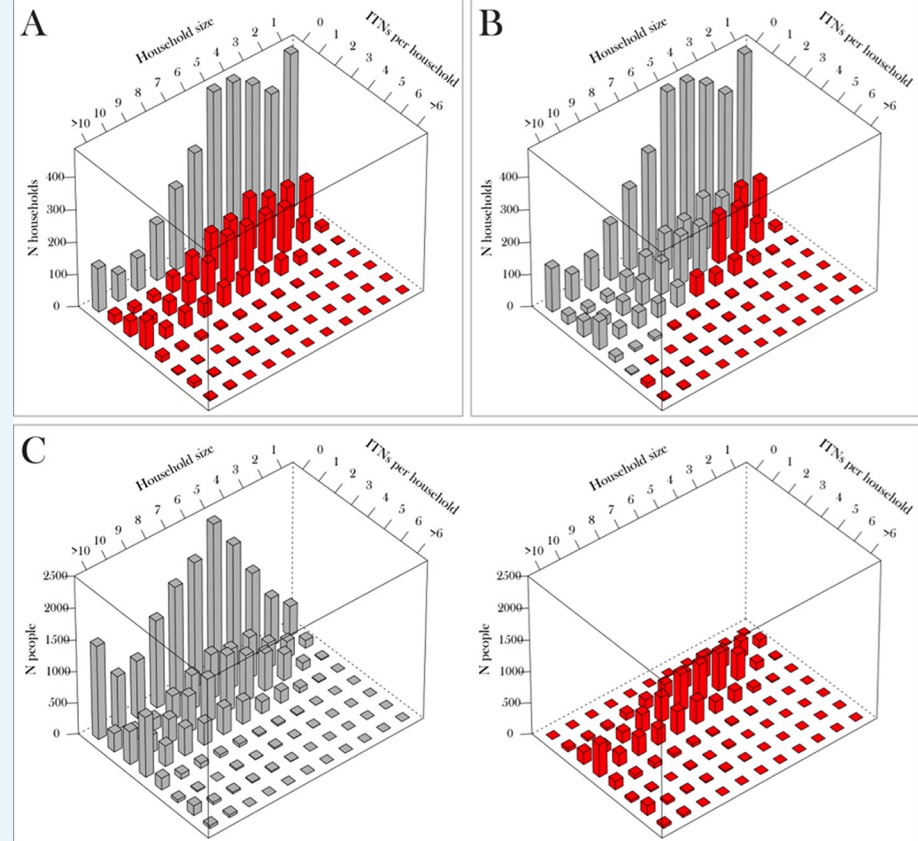

**Appendix figure 6**. Calculation of indicators 1–3 from two-dimensional histograms for Nigeria 2010 Malaria Indicator Survey. (**A**) *% HHs with one or more net,* obtained by dividing the sum of red bars (HHs with one or more ITN) by the sum of all bars. (**B**) *% HHs owning enough nets for one-between-two*, obtained by dividing sum of red bars (i.e. those HHs meeting or exceeding the one-between-two criteria) by the sum of all bars. (**C**) *% population with access to ITN in the HH*, obtained by first converting bars to represent people rather than HHs, then dividing sum of individuals that would be able to access an ITN sharing with one other (red bars in right-hand histogram) with the sum of all individuals (grey bars in left-hand histogram). HHs, households; ITNs, insecticide-treated nets.

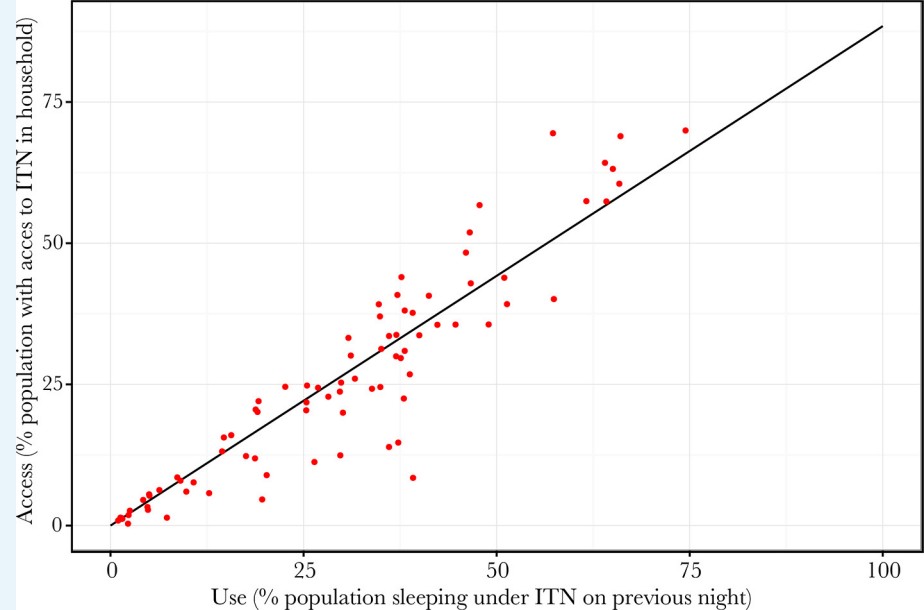

**Appendix figure 7**. The regression relationship between the proportion of people who slept under an ITN and the proportion of people with access to an ITN. ITN, insecticide-treated net.

