## [Decision Letter]

Thank you for submitting your work entitled "Coverage and system efficiencies of insecticide-treated nets in Africa from 2000 to 2017" for peer review at *eLife*. Your submission has been favorably evaluated by Prabhat Jha (Senior editor), Catherine Kyobutungi (Reviewing editor) and two reviewers, one of whom has agreed to reveal her identity: Eline Korenromp.

The reviewers have discussed the reviews with one another and the Reviewing editor has drafted this decision to help you prepare a revised submission.

Summary:

The authors, using data from 102 national surveys, triangulated against delivery data and distribution reports, developed a Bayesian model to generate year-by-year estimates of four ITN coverage indicators: i)% households with at least one ITN; ii)% of children under five who slept under an ITN the previous night; iii)% household with at least one ITN for every two people; and iv)% of the population with access to an ITN within their household.

The authors also explored the impact of two potential 'inefficiencies': uneven net distribution among households and rapid rates of net loss from households. The findings show worsening of over-allocation (a measure of inefficiency) over time as overall net provision has increased, and more rapid ITN loss from households than previously thought. This has implications on the universal coverage (only 77%) that can be expected from the current estimate of 920 million additional ITNs that have been computed as the need through 2017. By improving efficiency, however, the 920 million ITNs could yield population access as high as 95%.

The study presents an important situation analysis, of progress in distribution and resulting protective coverage with ITNs, the primary malaria prevention strategy in most of the high-malaria-burden countries in Africa. The dynamic model developed for this purpose is elegant, building on and refining earlier analysis, and adds value by producing more credible and considerably different estimates of the extent that ITNs are mis-allocated to some households, and the high rates of loss of ITNs from households – with the resulting inefficiencies varying among countries. The up-to-date estimates of ITN use will be useful to other researchers. Building on similar previous work, the authors have developed a model which allows them to break down coverage into estimates of multiple indicators that take account of the variation in household size, and this is an important advance on previous work. The paper also has projections of the coverage achieved if distribution can be done more efficiently or if people retain their nets for longer – this is also novel.

Essential revisions:

The areas of concern include the following:

1) The absence of concrete suggestions on how national malaria control programs could aim to reduce the identified inefficiencies, specifically over-allocation of ITNs. The authors should give examples of the operational strategies for better targeting and/or pre-screening of households' current ITN situation, and indicate which of these has proven to work, in what settings.

2) Different choices could have been made in the modelling, and this makes some of the conclusions more uncertain than what is currently presented. The authors should discuss these points and consider which outputs are most sensitive to the modelling choices. The estimates of the various indicators up to the present might not be much affected, but the projections forwards would be.

3) The net loss function (the rate at which people discard nets) is estimated by combining the numbers of nets entering a country, the number reported as distributed by control programmes, and the numbers owned by households. Flaxman et al. found that nets were retained were much longer than would be expected from matching up the distribution data and coverage data from cross-sectional surveys and as a result they introduced bias parameters to account for the discrepancy between data sources. It is not clear how this was handled in the modelling. The authors should describe this aspect of the methodology and the implications on the findings.

Minor points:

The estimated sharp fluctuations over time in ITN coverage indicators in some countries: are these real (if periodic campaigns dominate distributions?), or reflecting missing or incomplete data, e.g. in NMCP-reported distributions? In particular, comment on the odd time patterns in DRC and Ghana and/or a few other countries with irregular patterns.

[Editors' note: further revisions were requested prior to acceptance, as described below.]

Thank you for resubmitting your work entitled "Coverage and system efficiencies of insecticide-treated nets in Africa from 2000 to 2017" for further consideration at *eLife*. Your revised article has been favorably evaluated by Prabhat Jha (Senior editor), a Reviewing editor, and two reviewers. The manuscript has been improved but there are some remaining issues that need to be addressed before acceptance, as outlined below:

The reviewers still have concerns about the conclusions on net retention. The model in this paper only estimates ITN retention indirectly while the six studies that Flaxman depended on were directly measuring it. The bias parameters of Flaxman et al. were introduced because of a discrepancy between the different sources of data, if a loss function of longer duration is used. This discrepancy could be due to systematic differences between the data sources, and so the reviewers feel that the conclusion of the present paper that nets are retained for a shorter time than was previously thought is less certain than the authors suggest. This also has implications for the point about sharp fluctuations in the predicted ITN coverage over time, which presumably would be smoother with a loss function of longer duration.

We recommend that the authors conduct a sensitivity analysis that assumes longer retention of nets, showing how any projections change. In any case this uncertainty should be acknowledged and the implications for the projections discussed.

In addition, net loss and retention consider the physical presence of the ITNs in households, but not their effectiveness in repelling mosquitoes. According to manufacturer and WHOPES evidence and guidance on ITN's effective durability, is it justified to assume a median 36-month retention (in the optimized scenario), or does this imply a proportion of ITNs modelled as effective for longer than is realistically assumed? The authors should expound on this point in the discussion – for LLINs specifically

For the country-level estimates: Are the 4 international indicators (ownership, access, usage) calculated and shown for each country's overall national population (including sub-populations at no risk, just as the DHS, MICS and MIS surveys) or for the sub-population at risk of malaria? What assumptions, if any, are made about (mis-)allocation of ITNs to populations not at risk of malaria? Please clarify this in the appropriate section of the manuscript

Minor suggested edits:

If the world limit allows, please define 'Population access' in the Abstract.

Paragraph three, subsection “Net Loss”, 'minimised over-allocation': From the annex (and subsection “Estimating ITN Requirements to Achieve Universal Access”) it seems that this is ZERO over-allocation; please qualify or rephrase accordingly.

Figure 10:

a) Parts of the diagram are confusing or not clear – please correct as needed:b) An arrow (or, equality sign) is missing from HH-level ITN ownership to the box below (National total ITNs in HH?). c) An arrow missing from National net crop to, Observed bivariate histogramsd) National net crop equals National total ITNs in HH 5 boxes below, but the equality is not indicated.

Appendix 1: The scaling across countries to get to continent-totals is not very clear.

---

## [Author Response]

*Essential revisions:*

*The areas of concern include the following:*

1) The absence of concrete suggestions on how national malaria control programs could aim to reduce the identified inefficiencies, specifically over-allocation of ITNs. The authors should give examples of the operational strategies for better targeting and/or pre-screening of households' current ITN situation, and indicate which of these has proven to work, in what settings.

In this work we sought simply to identify and measure the magnitude of over-allocation and under-retention that exists and the changes in these factors over time. We felt this was a vital initial step and, of course, agree that it now paves the way for subsequent work to explore strategies for better targeting. We would propose, though, that this substantial body of new work would form one (or likely many) new studies in their own right, and would not be appropriately placed within the current submission which already covers a lot of new ground.

Solving the over allocation problem from an operational standpoint is something currently being discussed in policy fora. In the early years of integrated campaigns nets were allocated to a specific target group (e.g. children under five), but since the shift of policy to universal coverage this has become more difficult. Currently the distribution of nets requires a registration process within a county in order to identify beneficiaries thereby leaving large scope for under- and over-allocation. Work has been done at the Global Fund to strengthen the registration process but they have yet to identify concrete solutions that work at scale. Mozambique, Burundi and Madagascar have implemented systems based on village hierarchy that have been partially effective at small scales, but these do not lend to implementations at the scale of our analysis (country or continent level). Increasingly it is thought that solutions to the problem of over allocation need to be addressed in the planning stages rather than post hoc.

Given these complications, it is clear that defining strategies to address over-allocation in the field will remain a challenge – one to which modelling may be able to make a contribution alongside those with expertise of the reality of the logistics on the ground. For now, however, we maintain that by identifying that the phenomenon of over-allocation exists, is widespread, and contributing substantially to lower coverage yields, our study makes an important first step in motivating future work to improve distribution systems in-country.

*2) Different choices could have been made in the modelling, and this makes some of the conclusions more uncertain than what is currently presented. The authors should discuss these points and consider which outputs are most sensitive to the modelling choices. The estimates of the various indicators up to the present might not be much affected, but the projections forwards would be.*

We thank the reviewers for this comment. We felt that, because of the policy nature of this paper, most modelling assumptions were better described in full in the Appendix. For every component of the model we provide a conceptual and formal distribution of the model that outlines all the assumptions made and justifications for why we made them. As the reviewers expressed concern for the projection assumptions, we have added additional text to the assumptions section for future projections in the Appendix (Future predictions).

*3) The net loss function (the rate at which people discard nets) is estimated by combining the numbers of nets entering a country, the number reported as distributed by control programmes, and the numbers owned by households. Flaxman et al. found that nets were retained were much longer than would be expected from matching up the distribution data and coverage data from cross-sectional surveys and as a result they introduced bias parameters to account for the discrepancy between data sources. It is not clear how this was handled in the modelling. The authors should describe this aspect of the methodology and the implications on the findings.*

The Flaxman et al. paper was the inspiration for this work. However, there were several fundamental problems with their work. Their loss function was constant over a given year with very minor reductions for subsequent years until all nets are discarded at once after three years (see brown line Figure 7). This function is almost piece-wise linear and does not resonate with available evidence on net retention from field settings (e.g. see Killian et al.’s Net Calc). This limitation of their loss function was compensated for by introducing a ‘bias’ parameter that helped triangulate the entire modelling chain. As a result of this statistical compensation we do not believe their loss function represented reality particularly well – it is inconceivable that everyone in a given country discards all their nets at once at the three year mark. Additionally the Flaxman et al. net loss was fitted from 6 studies outside the main compartment model and fixed, whereas we learn a data driven, time varying loss function directly from the survey, manufacturer and NMCP data. We have discussed these limitations in the Methods section.

Minor points:

The estimated sharp fluctuations over time in ITN coverage indicators in some countries: are these real (if periodic campaigns dominate distributions?), or reflecting missing or incomplete data, e.g. in NMCP-reported distributions? In particular, comment on the odd time patterns in DRC and Ghana and/or a few other countries with irregular patterns.

We believe these fluctuations to be real. The priors on distributions are set as uniform and therefore in the absence of concrete data driving these sharp fluctuations the model will prefer smoother distributions. Sharp fluctuations only arise when triangulation between the NMCP, manufacturer and survey data demand it. In the absence of data the model will choose a more parsimonious and even distribution pattern. It should also be noted that these sharp distributions are accompanied by full Bayesian credible intervals capturing our uncertainty about the distribution pattern (if that uncertainty is there).

[Editors' note: further revisions were requested prior to acceptance, as described below.]

*The reviewers still have concerns about the conclusions on net retention. The model in this paper only estimates ITN retention indirectly while the six studies that Flaxman depended on were directly measuring it. The bias parameters of Flaxman et al. were introduced because of a discrepancy between the different sources of data, if a loss function of longer duration is used. This discrepancy could be due to systematic differences between the data sources, and so the reviewers feel that the conclusion of the present paper that nets are retained for a shorter time than was previously thought is less certain than the authors suggest.*

The key point of contention here revolves around how the NMCP distribution data triangulates with the loss function. We disagree that the measurement of the loss function, as modelled in Flaxman et al., is suitable and disagree that the bias parameter allows a loss function of longer duration. Below we explain the shortcomings of the Flaxman et al. approach and the true purpose of the bias parameter.

There are two modelling choices that can be made here:

1) The approach of Flaxman et al. – use a loss function based on 6 in field studies, and adjust the NMCP distribution data for a hypothesised systematic bias.

2) Our approach – learn a highly flexible loss function directly from the data and only adjust the NMCP distribution data for random noise.

To convince you that our approach is inherently a less subjective modelling choice consider the two following issues with the Flaxman et al. approach:

a) Net loss function:

Flaxman et al. paramaterised their loss function based on 6 studies with differing time points under no standardised conditions between studies. In these highly controlled studies key factors of net loss predominantly driven by living conditions, household behaviour and attitudes (Killian et al., 2015) are unaccounted for by the very nature of the study. Additionally some of the studies used in Flaxman et al. are retrospective studies where recall bias was a significant and unaccounted for confounder (Killian et al., 2015). We also must stress again just how implausible the loss function resulting from these studies is – the function is essentially constant (reductions with an average of 5% per year) for 3 years and then a complete stepwise reduction to zero at the 3-year mark. Under this loss function, using the upper 95% confidence interval, at most only 30% of nets can be lost after 3 years; immediately after which all nets are discarded. We believe that the implausibility of this loss function highlights the inconsistencies in paramaterising a function based on a small sample of unstandardized studies.

In contrast to this approach, we learn the loss function directly from the data, imposing an agnostic uninformative prior on the underlying half-life and duration. In our approach nets are lost smoothly over time (with no sudden discards) and the model has the choice of a long or short half-life depending the country and the year. From our fitted function (Figure 7) we find large variability between countries that would not be able to be represented using the Flaxman et al. loss function. Additionally, we actually say the duration of nets is longer than what Flaxman et al. say, with nets lasting up to 5 years on average (Figure 7 red curve). The key disagreement between our fitted loss function and that of Flaxman et al. is the year-to-year rate of reduction, and that all nets are lost exactly on the three year mark.

To further assure the reviewers of the implausibility of Flaxman et al. loss function, consider looking at the raw household survey data and extracting information on net age directly from each survey. When averaged across all surveys used in our study the average proportion of the total number of nets that were:

Less than one year old was 57%,

Between one and two years old was 20%

Between two and three years old was 5%

Greater than three years old was 18%

Obviously there are a lot more complexities to these simple summaries, including when the survey was sampled, the distribution pattern, recall bias etc. But one thing is clear, the year-to- year loss rate is very non-linear and probably much greater than 5% (which Flaxman et al. claim), and a large number of nets are older than 3 years (which the Flaxman et al. does not allow). This once again calls into question the plausibility of the Flaxman et al. loss function, but is consistent with our loss function.

b) The bias parameter:

The bias parameter in Flaxman et al. is used to systematically adjust the NMCP distribution data to match the household survey data. This is essential in their approach as given their constrained loss function (described above) the survey data would not match the distribution data without some form of adjustment. There are several conceptual and implementation problems with this bias parameter. Firstly this bias parameter is a very poor linear fit (see Figure 5, Supplementary information Flaxman et al.), and does not account for the sampling time of the survey or the recall bias inherent in the survey questions. The poor linear fit suggests substantial country specific variation. Second the prior does not model country specific variation making the assumption that all NMCP distribution data is biased in the same way.

Conceptually the problem with the bias parameter is a reliance on the assumption: *the NMCP data is systematically biased (constant) across all countries and years*. Flaxman et al. provide no justification for this assumption, and we believe it to be very unlikely that all NMCP data, reported from different countries, in different years are systematically biased in the same way. This is the equivalent to saying that for every country the number of nets reported to be distributed should be scaled by the *same* constant.

To further prove to the reviewers that using a bias parameter is not a sound modelling choice to explain the data, consider a like for like re-run of the Flaxman et al. bias parameter analysis. We take the weighted survey data totals for the survey questions:

Is your net less than one year old?

Is your net between 1 to 2 years old?

Is your net between 2 to 3 years old?

And regress these (on a logarithmic scale) against the corresponding NMCP distribution results in that year (with linear interpolation to match the average year of the survey). If the Flaxman et al. hypothesis of the NMCP data being systematically biased holds the intercepts of the three linear models (in Figure 16 where the intercepts are in the brackets, the regression bias line is in green and the one-to-one line is in red) should be the same. However, these intercepts are orders of magnitude different (note the logarithmic scale) suggesting steep losses of nets and no systematic bias. A far more parsimonious explanation is that the differences in the number of LLINs from the Survey and NMCP are explained by the loss of nets occurring through time. This is supported by the increasing bias parameter, as nets get older. Indeed this analysis suggests that the sole purpose of Flaxman et al. bias parameter is to compensate for the deficiencies in their poorly characterised loss function.

Author response image 1.**DOI:**
http://dx.doi.org/10.7554/eLife.09672.019

In contrast we do not assume any systematic bias on the NMCP data, we only assume that the NMCP data contains random noise and let the variation be described by a flexible loss function.

Therefore in summary we hope we have made clear that opting for modelling approach (2) over (1) is a considerably more parsimonious choice making far fewer assumptions and letting the data speak. We therefore have high confidence in the fits of our modelled loss functions. We stress that we have been completely transparent with the modelling assumptions in the Appendix, and have made very few subjective decisions and focused on a data driven approach.

*This also has implications for the point about sharp fluctuations in the predicted ITN coverage over time, which presumably would be smoother with a loss function of longer duration.*

We believe the reviewers are mistaken that the cause of sharp fluctuations in ITN coverage is due to the loss function. The predominant and overwhelming cause of these fluctuations is the delivery and distribution data, not the loss function. Sharp fluctuations arise due to real, large distributions of nets captured in both the distribution and survey data. See a typical example of this below for Togo.

Author response image 2.**DOI:**
http://dx.doi.org/10.7554/eLife.09672.020

In Figure 17, green dots are the manufacturer deliveries, purple triangles are the NMCP distributions, red triangles are the house-hold surveys, the red line is the number of LLINS, the yellow line is the number of cITNs and the black line is the residual stock. The fluctuations are caused by real large deliveries and distributions causing a saw tooth pattern in the number of nets. The reason the number of nets does not increase exponentially with the deliveries is due to loss, but this loss is smooth over time. Using the Flaxman et al. loss function would cause far less smoothness due to restriction of all nets being discarded at the three-year mark. To highlight this consider what would happen in the above plot if all nets were discarded after 3 years. As highlighted in the previous rebuttal the distribution of NMCP ITNs is uniform intra- years, as nothing in the data suggests a non-uniform pattern, thereby implicitly choosing a smooth distribution pattern unless the data would suggest otherwise.

We recommend that the authors conduct a sensitivity analysis that assumes longer retention of nets, showing how any projections change. In any case this uncertainty should be acknowledged and the implications for the projections discussed.

Given the above discussion about the suitability of our loss function we feel the reviewers will agree that a sensitivity analysis is unnecessary given our framework. If we used a loss function such as that in Flaxman et al., this sensitivity analysis would be warranted to test the validity of using such a constrained loss function. However, given the data driven, assumption free nature, of our analysis this is unnecessary as the sensitivity analysis is implicitly contained within the Bayesian framework. That is, during every fit, we explore a full range of candidate functions from differing retention times to sharp and smooth transitions and average over these to find the best function that fits the data. Therefore, during every fit, we perform a statistically robust sensitivity analysis. To state this differently our model allows for a loss function similar to that of Flaxman et al., but this loss function is not supported by the data at all.

That said, one of the important results of this paper is exploring the effects of a longer retention net half-life (3 year half-life). These results are presented and discussed in the main paper (Figure 8 red line) and in Appendix 1.

*In addition, net loss and retention consider the physical presence of the ITNs in households, but not their effectiveness in repelling mosquitoes. According to manufacturer and WHOPES evidence and guidance on ITN's effective durability, is it justified to assume a median 36-month retention (in the optimized scenario), or does this imply a proportion of ITNs modelled as effective for longer than is realistically assumed? The authors should expound on this point in the discussion – for LLINs specifically*

The scope of this paper was restricted to ownership and usage indicators and not on the effectiveness of nets once they are being used. Of course the insecticidal and barrier effectiveness of a nets diminish in a complex multifaceted way through time but due to these complexities, we did not expand our analysis to effectiveness and focus on the standard RBM definitions of ownership and use. We have expanded on this point in the Discussion.

For the country-level estimates: Are the 4 international indicators (ownership, access, usage) calculated and shown for each country's overall national population (including sub-populations at no risk, just as the DHS, MICS and MIS surveys) or for the sub-population at risk of malaria? What assumptions, if any, are made about (mis-)allocation of ITNs to populations not at risk of malaria? Please clarify this in the appropriate section of the manuscript

We use population at risk as the denominator in all the analysis in this paper and in doing so we avoid the problem of biased estimates by including populations not at risk. This however does make the assumption that there is no allocation of nets outside populations at risk. We have added a line in the main manuscript explicitly stating this assumption and thank the reviewers for pointing out this omission. However it is also vital to recognise that this adjustment by populations of risk only applies to a handful of countries in sub-Saharan Africa (e.g. Ethiopia, Kenya etc.) as all others are universally at risk.

*Minor suggested edits:*

*If the world limit allows, please define 'Population access' in the Abstract.*

These terms are fully described in the Introduction, subsection “Modelling coverage”.

*Paragraph three, subsection “Net Loss”, 'minimised over-allocation': From the annex (and subsection “Estimating ITN Requirements to Achieve Universal Access”) it seems that this is ZERO over-allocation; please qualify or rephrase accordingly.*

We thank the reviewers for this suggestion and have clarified this statement in the Methods and Appendix.

*Figure 10:*

a) Parts of the diagram are confusing or not clear – please correct as needed:b) An arrow (or, equality sign) is missing from HH-level ITN ownership to the box below (National total ITNs in HH?). c) An arrow missing from National net crop to, Observed bivariate histogramsd) National net crop equals National total ITNs in HH 5 boxes below, but the equality is not indicated.

*Appendix 1: The scaling across countries to get to continent-totals is not very clear.*

Figure 2 describes the Bayesian compartment model only, we choose not to tie together the Poisson model for the indicators as felt this would make the figure too complex.